# Hydrological impacts of global land cover change and human water use

Joyce H.C. Bosmans[1], L.P.H. (Rens) van Beek[1], Edwin H. Sutanudjaja[1], and Marc F.P. Bierkens[1,2]

[1]Department of Physical Geography, Faculty of Geoscience, Utrecht University, the Netherlands
[2]Unit Soil and Groundwater Systems, Deltares, Utrecht, the Netherlands

*Correspondence to:* Joyce Bosmans (J.H.C.Bosmans@uu.nl)

**Abstract.** Human impacts on global terrestrial hydrology have been accelerating during the $20^{th}$ century. These human impacts include the effects of reservoir building and human water use, as well as land cover change. To date, many global studies have focussed on human water use, but only a few focus on or include the impact of land cover change. Here we use PCR-GLOBWB, a combined global hydrological and water resources model, to assess the impacts of land cover change as well as human water
use globally in different climatic zones. Our results show that land cover change has a strong effect on the global hydrological cycle, on the same order of magnitude as the effect of human water use (applying irrigation, abstracting water for e.g. industrial use, including reservoirs etc). Globally averaged, changing the land cover from 1850 to that of 2000 increases discharge through reduced evapotranspiration. The effect of land cover change shows large spatial variability in magnitude and sign of change depending on e.g. the specific land cover change and climate zone. Overall, land cover effects on evapotranspiration are largest
for the transition of tall natural vegetation to crops in energy-limited equatorial and warm temperate regions. In contrast, the inclusion of irrigation, water abstraction and reservoirs reduces global discharge through enhanced evaporation over irrigated areas and reservoirs as well as through water consumption. Hence in some areas land cover change and water distribution both reduce discharge, while in other areas the effects may partly cancel out. The relative importance of both types of impacts varies spatially across climatic zones. From this study we conclude that land cover change needs to be considered when studying
anthropogenic impacts on water resources.

## 1 Introduction

The anthropogenic impact on the global terrestrial hydrological cycle has many aspects. Both emission-driven climate change as well as more direct human interventions such as dam building and water withdrawals (for domestic, industrial and agricultural use, including irrigation) have a strong impact on future water availability, floods and droughts (e.g. Hirabayashi et al.,
2013; Haddeland et al., 2014; Wanders and Wada, 2015; Winsemius et al., 2016; Veldkamp et al., 2017). Additionally, humans have altered a large part of the land surface, replacing 33% (Vitousek et al., 1997) or even 41% (Sterling et al., 2013) of natural vegetation by anthropogenic land cover such as crop fields or pasture. Such land cover change can affect terrestrial hydrology by changing the evaporation to runoff ratio. To date, few studies focus on land cover change when assessing the anthropogenic impact on the global terrestrial hydrological cycle. Here, we compare the effects of land cover change, mainly the expansion of

crop and pasture at the expense of natural vegetation, to human water use, i.e. water abstraction for irrigation and non-irrigation use as well as reservoir building. We compare these effects globally as well as spatially, providing an in-depth analysis across climatic zones.

Studies that have assessed the impact of land cover change on global terrestrial hydrology generally find decreased evapotranspiration and increased discharge. Comparing potential (i.e. natural) to actual (present-day) vegetation, Gordon et al. (2005) suggest that decreased evapotranspiration due to deforestation is larger than the increase in evapotranspiration due to irrigation, globally averaged. Piao et al. (2007) emphasize that the observed increase in runoff over the 20th century was not only due to climate change, but that land cover change was equally important, if not more important in some regions, based on experiments with the ORCHIDEE model. Using the LPJmL model, Rost et al. (2008b) report reduced evapotranspiration through reduction of transpiration and interception as natural vegetation is replaced by crops and pasture (grazing land). They furthermore report that the land cover change impact is larger than the climate change impact as well as the impact of water abstraction for irrigation, globally averaged (Rost et al., 2008b, a). Sterling et al. (2013) focus solely on land cover change, and like Rost et al. (2008b) find reduced evapotranspiration due to land cover change, with the conversion of natural vegetation to (rainfed) crops contributing more to the evapotranspiration reduction than the conversion to pasture, despite the latter affecting a larger area. Reduced evapotranspiration results in increased river discharge, albeit covering regional differences in magnitude and sign of change. On a regional scale, similar conclusions are reached by Haddeland et al. (2007) for North America and Asia, with the largest land cover induced changes in runoff occurring over South-East Asia. Hence, despite large variations amongst studies concerning the actual amount and spatial variation of evapotranspiration and runoff changes due to land cover change, related to e.g. uncertainties in evapotranspiration reconstructions, models and land cover maps (Boisier et al., 2014), land cover change is overall thought to have reduced global evapotranspiration and increased runoff to an extent that is at least of similar magnitude as the impact of climate change or other anthropogenic impacts such as irrigation.

In this study we investigate the impact of land cover change as well as human water use, providing a detailed analysis of changes in the water balance across the major climatic zones. Our objective is twofold: first we create new land cover parameter sets for 1850 and 2000 for the PCR-GLOBWB global hydrological model. Second, we use these parameters in sensitivity experiments to study the effect of land cover change in detail and compare to the effect of human water use (through e.g. irrigation, demand for industry, reservoirs), with an emphasis on annual mean river flow. A brief overview of experiments is given in Table 1. In addition to an in-depth analysis across climatic zones, this study adds to existing literature by introducing a novel land cover product and by using the global hydrological and water resource model PCR-GLOBWB (Van Beek et al., 2011; Wada et al., 2014; Dermody et al., 2014). Our land cover parameterization uses crop and pasture areas from the harmonized land use data by Hurtt et al. (2011), for historical years based on HYDE (Klein Goldewijk et al., 2011), who provide crop and pasture cover used in historical as well as future climate scenarios in CMIP5. We combine this with land cover type-specific parameters from GLCC (Global Land Cover Characterization, Olson 1994a, b) and MIRCA (Monthly Irrigated and Rainfed Crop Areas, Portmann et al. 2010). Land cover parameters are allowed to vary per land cover type as well as spatially. The methods of creating our land cover product are further detailed in Section 2, as is our experimental set-up. The resulting land cover change for 1850-2000 as well as its impact on global terrestrial hydrology are provided in Section 3, where land cover

impacts are furthermore compared to the impact of human water use (e.g. dams, irrigation). A discussion of our methods and results is given in Section 4, followed by conclusions in Section 5.

## 2 Methods

### 2.1 PCR-GLOBWB global hydrological model

5 Here we apply the PCRaster Global Hydrological Water Balance model, PCR-GLOBWB, at 0.5°x0.5° globally (roughly 50x50km). This global hydrological and water resources model includes the interaction between terrestrial water fluxes and human water use. It simulates the vertical water balance in two soil layers and an underlying groundwater layer, see Fig. 1. Water can be stored in the canopy, snow, soil, rivers, lakes, and groundwater. PCR-GLOBWB takes sub-grid variability into account by including soil type distribution (FAO Digital Soil Map of the World), the simulated fraction of area of saturated 10 soil (based on the Improved Arno Scheme, Todini 1996; Hagemann and Gates 2003) and the spatio-temporal distribution of groundwater depth based on the high resolution digital elevation model (as referenced by Van Beek et al. 2011) and the simulated groundwater storage. Several land cover types can be considered within one grid cell. These land cover types will be detailed in Section 2.2.

When human water use is included, irrigated crop fields receive additional water if precipitation and soil moisture alone do 15 not satisfy the crop demands. Paddy irrigated fields (rice) are covered by 5 cm of water during the growing season. Irrigation demand over non-paddy irrigated fields is computed by the model based on green water availability (evapotranspiration without irrigation) and the demand of the irrigated areas based on crop factors, see Van Beek et al. (2011); Wada et al. (2014) for details. Water demand for livestock, industry and domestic use is prescribed, using water demand estimates for 2000 from Wada et al. (2014) based on livestock densities, population densities and country-statistics on socio-economic development. Irrigation and 20 non-irrigation demand can be met by water from rivers, lakes, reservoirs and groundwater (Wada et al., 2011, 2014; De Graaf et al., 2014). Fossil groundwater abstraction is taken into account, which is the non-renewable part of groundwater abstraction not replenished by recharge. Fossil groundwater is a non-sustainable water source added to meet water demand, but it is not part of the active hydrological cycle (Wada et al., 2012; De Graaf et al., 2014). In order to limit abstraction, data sets on the relative contribution of surface and groundwater are used and a regional limit on pumping capacity is applied (Erkens and 25 Sutanudjaja, 2015). Furthermore, water can be lost through consumption, which is water abstracted for e.g. domestic, industrial and agricultural demand not returned to the hydrological cycle. For a more detailed model description, see Fig. 1 and Van Beek et al. (2011); Wada et al. (2014).

We force each model experiment with combined ERA-Interim and CRU-TS3.21 temperature, precipitation and reference potential evapotranspiration from 1979-2010, thus providing 32 years of output for each experiment (following a spin-up of up 30 to 20 years). Reference potential evapotranspiration is computed using the FAO Penman-Monteith equation (Allen et al., 1998), and converted to vegetation specific potential evapotranspiration using crop factors (see below). The monthly temperature, precipitation and reference potential evapotranspiration are then broken down into daily values using ERA-Interim reanalysis (see e.g. Van Beek (2008); Sutanudjaja (2012) for the same method applied to CRU-TS2.1 and ERA-40).

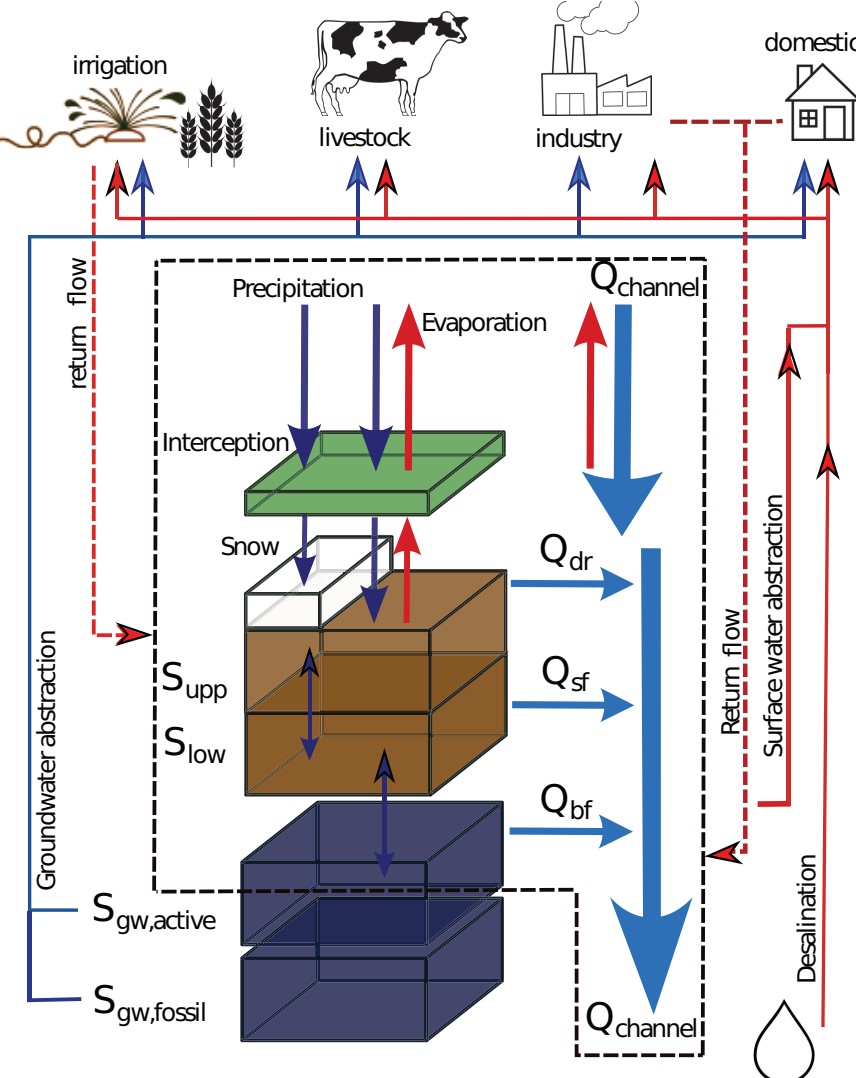

**Figure 1.** Overview of the PCRaster Global Water Balance model, PCR-GLOBWB. The vertical structure, within the black dashed lines, consists of canopy, two soil layers and a groundwater reservoir. Potential evapotranspiration is broken down into canopy transpiration and bare soil evaporation. Evaporation can occur from the canopy, depending on interception capacity and precipitation intensity, from the soil (depending on soil saturation). Transpiration depends on soil moisture and crop coefficients. Discharge along the channel network consists of direct runoff, interflow or subsurface flow and baseflow. In experiment HUM2000 water abstraction, irrigation and reservoirs are included, as is the use of desalinated water (Wada et al., 2014), hence all fluxes including those outside the black dashed lines are computed. Figure courtesy of S. Pessenteiner.

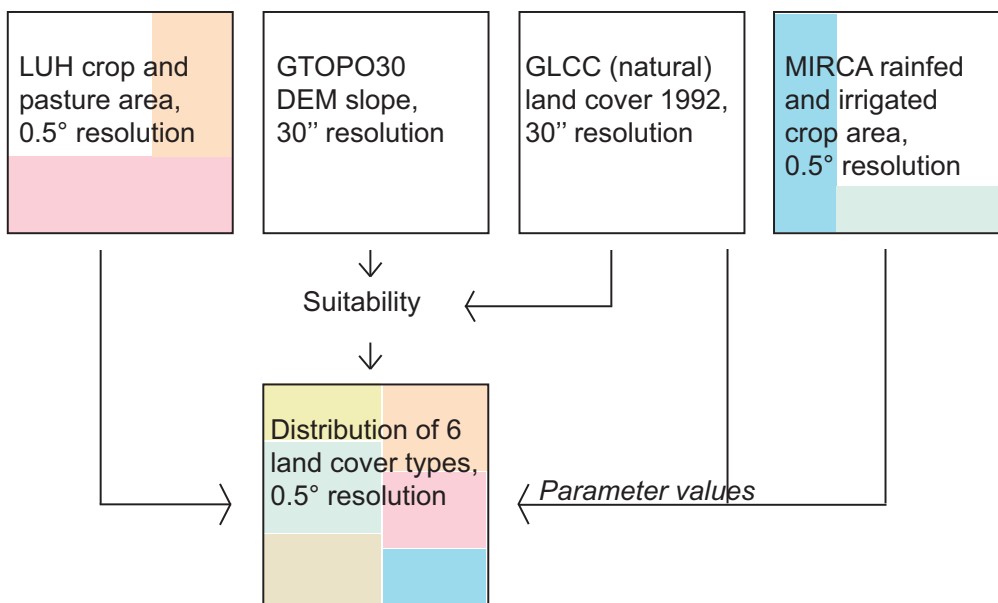

**Figure 2.** Schematic of how land cover parameters are constructed. Each block represents a 0.5°x0.5° grid cell. LUH refers to harmonized land use data from Hurtt et al. (2011), DEM refers to digital elevation map (Van Beek et al., 2011), GLCC is the Global Land Cover Characterization (Olson, 1994a, b; Hagemann et al., 1999) and MIRCA refers to Monthly Irrigated and Rainfed Crop Areas Portmann et al. (2010). After Dermody et al. (2014).

## 2.2 Land cover change

PCR-GLOBWB considers sub-grid variability in land cover by allowing for multiple land cover types per grid cell. Each land cover type is described by a different set of spatially and intra-annually varying parameter values, determining e.g. the amount of canopy interception, root depth etc. Here we include 6 land cover types: tall and short natural vegetation, pasture, and three

types of crops. Pasture covers a wide range of ecosystems, including intensive managed grasslands in for instance North West Europe as well as extensive rangeland similar to natural vegetation in drier parts of the world. Crops are separated into rainfed, non-paddy irrigated and paddy irrigated crops. We base the distinction between rainfed and irrigated crops on the MIRCA data set (Monthly Irrigated and Rainfed Crop Areas, Portmann et al. (2010)), and compute crop parameter values based on 26 spatially and temporally varying crop types. Including pasture and rainfed crops separately is an extension of previous PCR-

GLOBWB studies (e.g. Van Beek et al., 2011; Wada et al., 2011, 2014) as we focus on anthropogenic changes in land cover. We use fractional crop and pasture cover for 1850 and 2000 provided by Hurtt et al. (2011) at 0.5°x0.5° resolution. The data by Hurtt et al. (2011) extend to 2100 per Representative Concentration Pathway, allowing us to include land cover change in later work focussing on anthropogenic impacts in the future. Other studies on land cover change are based on different sources. For instance, crop and / or pasture cover is often taken from Ramankutty and Foley (1999) instead of Hurtt et al. (2011) (e.g.

Piao et al., 2007; Rost et al., 2008b; Sterling et al., 2013).

Per land cover type and per grid cell, PCR-GLOBWB requires various parameters, such as the vegetation fraction per grid cell, the root depths for the improved Arno Scheme, the crop factor to determine the land cover-specific potential evapotranspiration and the interception capacity to partition precipitation into interception and throughfall. As there is no direct source of information on these parameters for historical (or future) land cover changes, we combine available data sets following the approach of Dermody et al. (2014), see Fig. 2. To identify which types of vegetation actually exist per grid cell per land cover type we first create a suitability map using the Global Land Cover Characterization (GLCC, (Olson, 1994a, b; Hagemann et al., 1999)) as well as the slope based on GTOPO30 digital elevation model at 30" (arcsec, roughly 1km x 1km, Van Beek et al. (2011)). Suitability is deemed highest in areas presently covered by crop or pasture according to GLCC, within which suitability decreases with increasing slope. Outside these areas, suitability further decreases with distance to these areas as well as with increasing slope. The suitability is used iteratively to select the most suitable cells until the area required by Hurtt et al. (2011) for either 1850 or 2000 was met, first for crops and then for pasture. The remaining area, not filled with crop or pasture, is filled with reconstructed natural vegetation from the GLCC dataset (tall or short, based on the forest fraction). The resulting 30" information is then combined to the effective land cover parameter values per land cover type per grid cell at 0.5°x0.5° by taking the average of the GLCC parameter values over the grid cell area for natural vegetation or pasture and filling in the crop area using MIRCA input. Note that by moving from the 0.5°x0.5° model resolution to the 30" resolution of GLCC and GTOPO we allow for different vegetation types, and therefore potentially different parameter values, to be included in the natural and pasture land cover types over time. The grid cell and land cover type specific parameter values thus reflect a mixture of crop, pasture or natural vegetation types. As an example, the spread of crop factors is given in appendix Fig. A1, as are the maximum crop factors in Fig. A2. The spread represents the variation over space and time, e.g. higher crop factors occur during the growing season. Figures A3 and A4 show the root distribution in the two modeled soil layers. Crops, particularly irrigated crops, have root mainly in the upper soil layer, but for the other land cover types the root distribution varies spatially.

Note that we use the term land cover types, whereas especially pasture could also be considered as a land use type. However, by using global input from GLCC and MIRCA we do allow for the parameter values to vary spatially; e.g. a pasture field consisting of managed grassland will have different parameter values than a pasture field with shrubs or savanna. A table of GLCC ecosystems classified as pasture in experiment LC2000 is available in the supplementary materials (Area_table_pasture.tbl). Tall natural vegetation can represent dense forest, but also savanna or shrubs. Rainfed and non-paddy irrigated crops also vary spatially depending on which crops grow where according to MIRCA. Therefore within the 6 land cover types we represent a larger variety of vegetation types, as opposed to studies that use for instance plant functional types (PFTs) which typically do not have spatial variability in the PFT characteristics (albeit allowing for different PFT combinations in different grid cells).

## 2.3 Experiments

To test the sensitivity of global terrestrial surface hydrology to land cover change we perform two experiments with exactly the same model version and boundary conditions, except for the land cover: LC1850 and LC2000. Changes in vegetation cover per land cover type are shown in Fig. 4 and are briefly described in Section 3.1. Note that human water use (applying irrigation,

reservoirs, abstracting water for e.g. industrial use) is not taken into account in LC1850 or LC2000, so essentially only the model core in the black dashes in Fig. 1 is used and all crops are rainfed.

Furthermore, we repeated the LC2000 experiment but with human water use (HUM2000), so this experiment includes water withdrawals, reservoirs and the application of irrigation to the paddy and non-paddy irrigated land cover types (Fig. 1). Water

demands for industry, domestic and livestock, water delivery from desalinization, and reservoirs are fixed for the year 2000 based on those used in Wada et al. (2014). Paddy and non-paddy irrigated areas are also fixed, as the land cover parameters in our experiments do not include interannual variability. These experiments should therefore be viewed as idealized sensitivity experiments, set up to study the direct impacts of land cover change and human water use separately and combined.

Using these three experiments (see Table 1) we can test how the sensitivity to land cover change compares to the sensitivity

to human water use, i.e. comparing LC2000 to LC1850 as well as HUM2000 to LC2000. For the combined effect we compare HUM2000 to LC1850 in selected figures. Note that we only change either the land cover (LC2000 vs LC1850) or the water use (HUM2000 vs LC2000), PCR-GLOBWB does not take into account precipitation and / or energy flux feedbacks.

## 2.4   Comparison to GRDC discharge

PCR-GLOBWB is a suitable tool to investigate the global hydrological cycle, as the model is set up to study the terrestrial water

cycle including the interaction with human water demand and use. Previous studies have shown that the model performs well compared to observations such as the Global Runoff Data Centre's (GRDC) discharge measurements, the Food and Agriculture Organisation's Aquastat product for water use and the total water storage of the Gravity Recovery and Climate Experiment (GRACE) (e.g. Wada et al., 2011, 2014). Here we present a brief comparison of discharge to 44 selected GRDC stations, representing the most downstream station of major rivers with catchment areas larger than 150,000 km$^2$. For the Amazon the

second-most downstream station is used as the most downstream one has a much larger catchment area compared to the other stations. The statistics are based on monthly discharge for the period in which each station has data available within 1979-2010 period.

Figure 3 shows that on average, PCR-GLOBWB over-estimates discharge compared to GRDC measurements in all three experiments. However, the R$^2$ values are high for each experiment, with (marginally) higher R$^2$ values for more realistic bound-

ary conditions (land cover of 2000 rather than 1850, including human water use). For these 44 stations, the combined average annual mean discharge for the periods in which GRDC data is available is 15618 km$^3$/yr for LC1850, 15828 for LC2000 and 15446 for HUM2000, compared to 13147 km$^3$/yr for the measurements. Thus the bias (model minus measurements) is 2471, 2681 and 2299 km$^3$/yr, respectively. Discharge in selected rivers per experiment is provided in Section 3.2.

The better fit of experiment HUM2000 becomes clearer when considering the root mean-square error (rmse) and the Kling-

Gupta Efficienty (KGE, (Gupta et al., 2009; López López et al., 2017)). The rmse is lower for HUM2000 compared to LC2000 for 33 out of 44 stations, and lower compared to LC1850 for 28 out of 44 stations. Similarly, the KGE is higher for HUM2000 in these stations (Figure 3). Experiments LC1850 and LC2000 perform very similar in comparison to GRDC measurements. Note however that our experiments are set up as idealized sensitivity experiments (see Section 2.3), and that while LC2000 has more crop and pasture cover representative of present-day, no irrigation is applied. Experiment HUM2000 does include

irrigation, as well as water use for other purposes, thus resulting in a better fit to measurements, despite keeping water demand and irrigation requirements fixed using values for the year 2000 (see Section 2.3).

## 3 Results

In this section we first describe the land cover change between LC2000 and LC1850 (Section 3.1). We then describe the impact
of land cover change on the terrestrial hydrological cycle and compare this to the impact of human water use, by looking at differences in the results of LC2000 vs LC1850 as well as HUM2000 vs LC2000. We use several analyses. In Section 3.2 we focus on the changes in the water balance, mainly discharge and evapotranspiration, showing global averages as well as grid cell specific changes averaged over the 32 year experiments. In Section 3.3 we use the subbasins defined in Section 3.1 to investigate how the hydrological cycle responds to specific land cover change in different climate zones. Last, in Section 3.4,
we show a Budyko plot for the 100 largest river basins to investigate whether changes in land cover or human water use shift the water partitioning between evapotranspiration and runoff within larger basins.

### 3.1 Land cover change

Figure 4 shows the change in land cover between 1850 and 2000 per land cover type. There is an overall reduction of tall and short vegetation to the advantage of pasture and crops, affecting all areas except high northern latitudes and the deep tropics
(Amazone and Congo). Overall, natural vegetation reduces by $34.8 \times 10^6 km^2$ between 1850 and 2000, roughly 26% of the total land surface. This is mostly taken over by pasture (increasing by $25.4 \times 10^6 km^2$, 19%) and rainfed crops (increasing by $7.9 \times 10^6 km^2$, 6%). The increase in irrigated area is about 1%, but irrigated areas will play a role in the HUM2000 experiment when surface evapotranspiration increases due to irrigation being applied. Note that in the land cover of 1850, some 10% of the area is already covered by crop or pasture, increasing to 36% in the 2000 land cover. The anthropogenic areas in 1850 are
mostly in eastern U.S. and western Europe, where some natural vegetation returns in the 2000 land cover (see Fig. 4).

For further analysis, we subdivided the world into subbasins, starting with subbasins larger than 30,000 km$^2$ (comparable in size to the Meuse basin in Europe or the Allegheny basin in the USA). Subbasins smaller than 30,000 km$^2$, mostly small endorheic or coastal basins covering only a few gridcells, were grouped. This resulted in 3995 subbasins, with a mean area of 33,396 km$^2$, ranging from 19.4 to 3,047,270 km$^2$. Within these subbasins a further division was made based on the dominant
land cover change (for instance mainly a reduction in tall natural vegetation and an increase in pasture, see Fig. A5) and the predominant Köppen-Geiger class, using the Köppen-Geiger classification of climatic zones of Kottek et al. (2006). Table 2 shows the areas in these subbasins. Most of the area within these subbasins experiences increased pasture cover at the expense of both tall and short natural vegetation (2000 minus 1850 land cover; green and red in Fig. A5, this also follows from Fig. 4). Conversion from tall natural vegetation to pasture is dominant in tropical South America, Africa as well as north and east
Australia (note that tall natural vegetation includes e.g. savannas and shrubs, as well as forests, see Section 2.2). Over mid-west North America, southern South America, southern Africa, the Arabian Peninsula, central Asia and south-west Australia the main land cover change is from short natural vegetation to pasture. Conversion from tall natural vegetation to crops affects

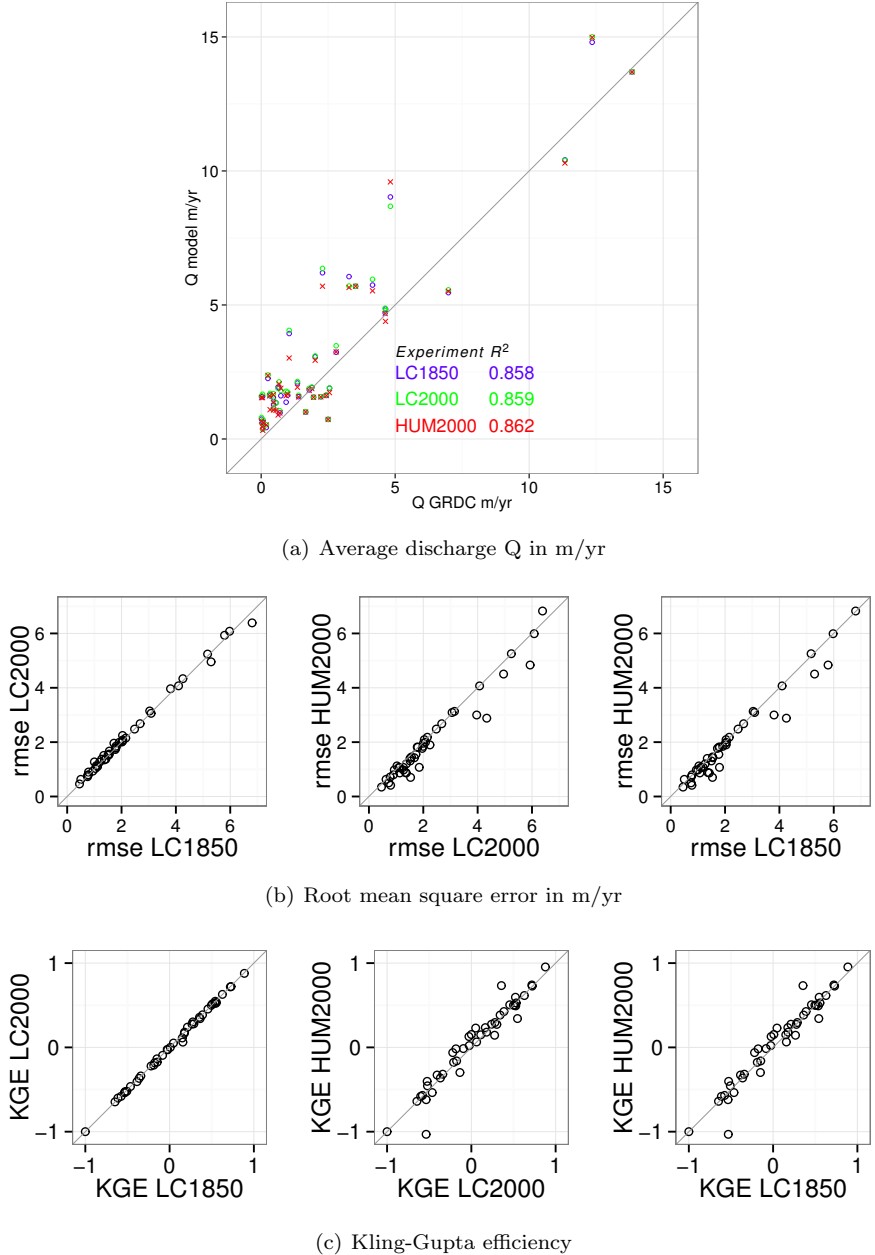

(a) Average discharge Q in m/yr

(b) Root mean square error in m/yr

(c) Kling-Gupta efficiency

**Figure 3.** Comparison of discharge computed by PCR-GLOBWB and measurements from GRDC stations. Figure (a) shows annual mean discharge in m/yr (discharge in km3/yr divided by catchment area in GRDC or on the model grid) of each experiment compared to observed discharge. Figure (b) shows root mean-square error (in m/yr), comparing the model experiments. A lower rmse indicates a better agreement to the GRDC data, which is the case for the HUM2000 experiment in the majority of stations. Figure (c) shows the Kling-Gupta efficiency (KGE). A higher KGE indicates a better agreement to the GRDC data, which is the case for the HUM2000 experiment. 44 stations were selected for this comparison (see table Comparison_GRDC.xlsx in supplementary materials). KGE values range between -1 and 1 except for one station (4103200 on Yukon River) where KGE is -3.1 for all experiments. For plotting purposes these values are set to -1 in Figure (c).

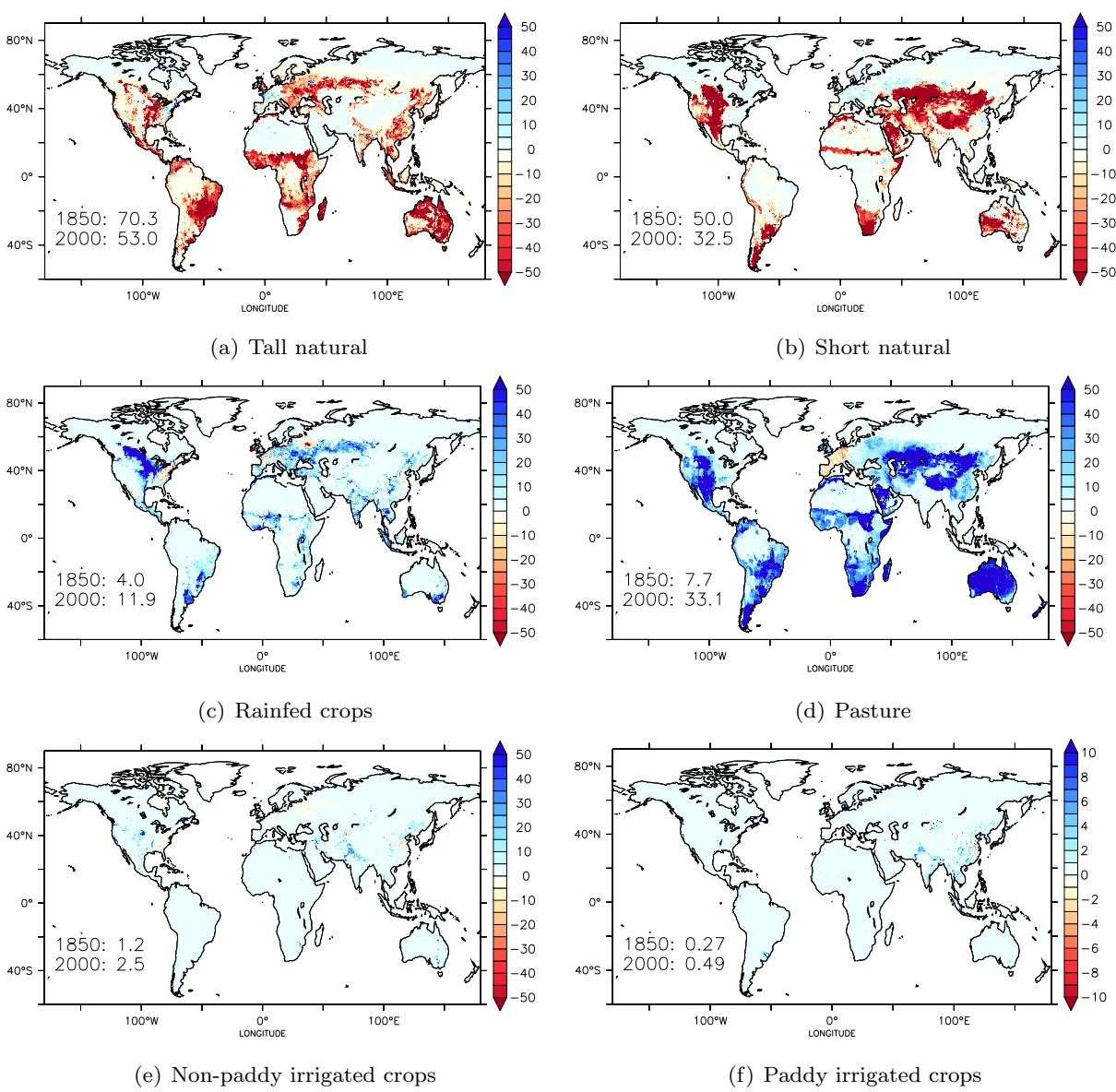

(a) Tall natural

(b) Short natural

(c) Rainfed crops

(d) Pasture

(e) Non-paddy irrigated crops

(f) Paddy irrigated crops

**Figure 4.** Changes in land cover in each of the 6 land cover classes, expressed in percentage of grid cells. Note that in figures a-e the scale reaches 50%, while in figure f, for paddy irrigated crops, it reaches 10%. The numbers indicate the surface area covered by a land cover type in 1850 and 2000 in $10^6$ km$^2$. Total land surface area in our experiments is $133 \times 10^6$ km$^2$ (Antarctica is excluded). In experiments LC1850 and LC2000 all crop fields are rainfed, only in HUM2000 do the paddy and non-paddy irrigated fields receive additional water.

mainly parts of central-eastern US, eastern Europe and south east Asia. In terms of climate zones, conversion of tall natural vegetation to pasture is the most dominant change in equatorial and warm temperate climates (Köppen classes A and C), while in arid and polar climates (B, E) the dominant change is from short to pasture. Conversion to crop is mainly from tall natural vegetation, most of which occurs in snow climates (Köppen class D). In total, in terms of area, 93% (123.7 km$^2$) of these
5 subbasins experiences at least some conversion from natural (tall or short) to anthropogenic land cover (pasture or crop). Only 2% is converted from anthropogenic back to natural vegetation, mostly in western Europe and eastern North America, and 5% experiences no land cover change at all ('Other' and 'noLC' in Table 2).

## 3.2 Changes in global hydrology: water balance

Figure 5(a) shows discharge changes due to land cover changes. Land cover changes can in- or decrease discharge, with
10 opposite changes occurring even within basins (e.g. Mississippi, Amazone). Global average annual mean discharge increases by 901 km$^3$/yr (LC2000 vs LC1850). This amounts to a 1.9% increase in global discharge. Discharge changes can reflect both local and upstream changes in land cover, the latter is clear for instance in the high northern latitudes where there is no land cover change (see Fig. 5(a)). Compared to the effect of human water use, land cover change effects are of similar magnitude but this global average masks a large spatial spread in sign and magnitude. Figure 5(b) shows that including human water use
reduces discharge in all affected rivers (HUM2000 vs LC2000), as a result of water being stored in reservoirs and abstracted for e.g. irrigation or industrial use. Blue areas in Fig. 5(b), where discharge increases, correspond to reservoirs, which are included in HUM2000 but not in LC2000. There is some variation in which rivers are more affected by the land cover change or the human water use, see Fig. 5. Table 3 shows discharge changes in 26 main rivers for all three experiments. 6 of the 26 rivers have decreased discharge due to land cover change, but this decrease is small compared to the impact of human water use. Also,
amongst these 6 is the Rhine where the overall of land cover change is a conversion of crop and pasture to natural vegetation, which on average decreases discharge. The large rivers in the tropics (Amazone and Congo) are not strongly affected by land cover change (Fig. 4) and therefore total discharge does not change (<1%, Table 3), although this masks some intra-basin in- and decreases. Of the 26 rivers in Table 3, discharge to the ocean from 17 rivers is more affected by human water use than land cover change. Globally averaged the reduction in discharge in HUM2000 compared to LC2000 is 1185 km$^3$/yr, which is
comparable in magnitude to the discharge increase due to land cover change (901 km$^3$/yr).

As the only difference between experiments LC2000 and LC1850 is in the land cover, the changes in discharge can be explained by differences in actET (actual evapotranspiration from the land surface, Fig. 6(a)). An increase in actET reduces discharge by removing water that would have gone into the rivers, and vice versa. Upstream regions of for instance the Dnieper and the Nile, where the relative increase in discharge due to land cover change is large, experience reduced actET. Globally
averaged, actET is reduced by 917 km$^3$/yr, or 1.6%, and total evapotranspiration (land surface evapotranspiration plus water-body evaporation) is reduced by 888 km$^3$/yr, or 1.5%. As expected, the increased discharge (901 km$^3$/yr) can almost fully be explained by changes in evapotranspiration, see Table 4. The remaining 13 km$^3$/yr is small (the total discharge per experiment is ∼47,000 km$^3$/yr, see Table 3) and can be attributed to small changes in total terrestrial water storage (lower by 17 km$^3$/yr in LC2000) and rounding errors in processing the model output.

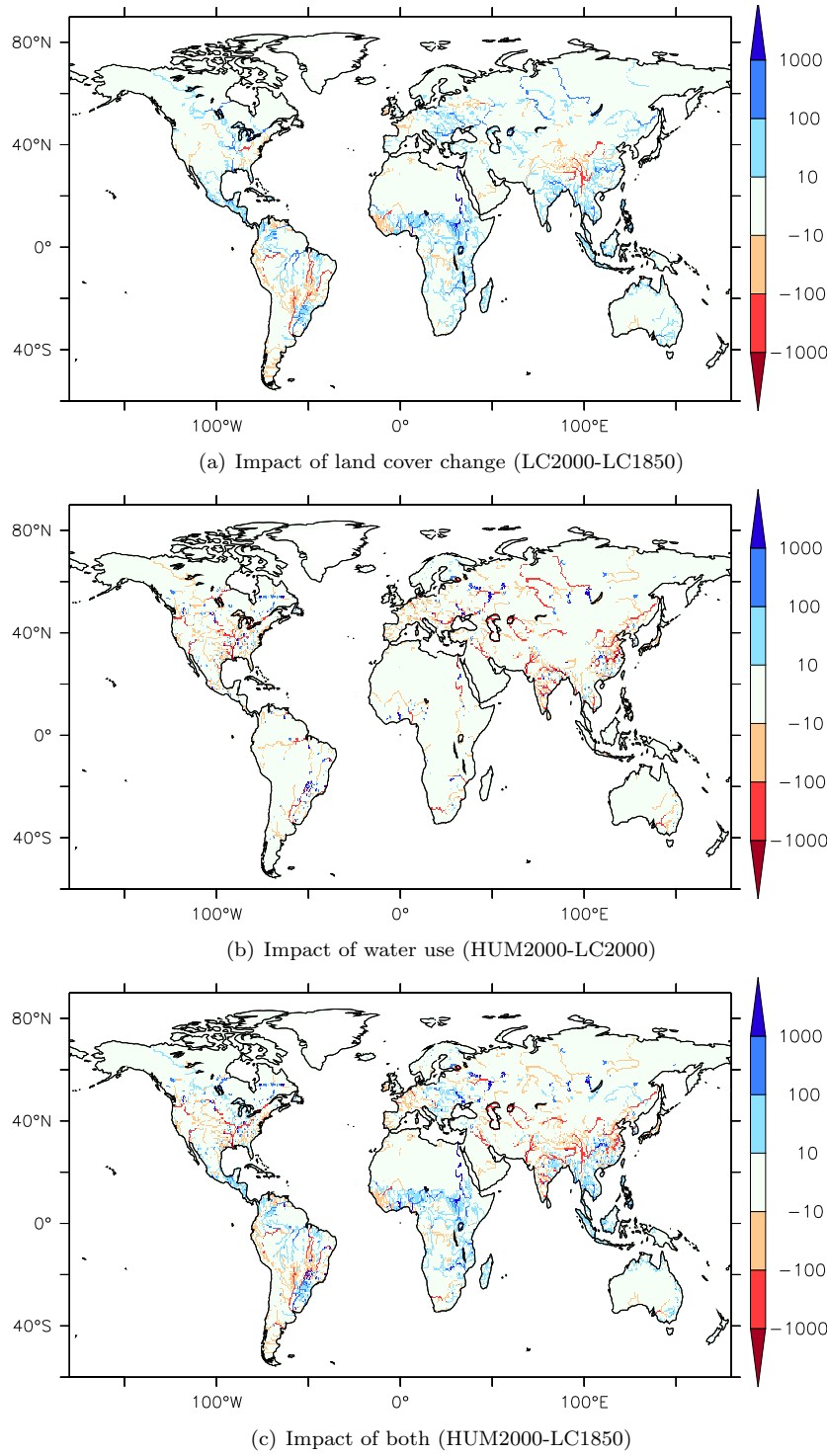

**Figure 5.** Difference in annual average discharge between the experiments, averaged over 1979-2010, in m$^3$/s. Higher discharge for HUM2000 occurs over reservoirs, which are not included in LC1850 or LC2000.

Most actET changes occur over eastern US, central America, south-east South America, tropical North Africa, central Europe and South-East Asia (Fig. 6(a)). These are areas with large land cover change (Fig. 4), but not all areas with large land cover change experience a strong change in actET. For instance, central US, southern Africa, central Asia (along $\sim$40°N) and Australia show little change in actET, despite strong changes in potential evapotranspiration due to land cover change (Fig. 7). These are generally water limited (potET > P), arid areas, where changes in potential evapotranspiration do not have a strong effect on actual evapotranspiration. In Section 3.3 we will further evaluate changes in different climate zones.

The effect of human water use on actET is slightly smaller than the effect of land cover change, as evapotranspiration is only increased over irrigated areas but shows a strong increase there (Fig. 6(b)). Globally averaged, evapotranspiration from the land surface is increased by 701 km$^3$/yr. Another 134 km$^3$/yr evaporates from water bodies, mainly the reservoirs. Total evapotranspiration is increased by 846 km$^3$/yr, which is not enough to balance the 1185 km$^3$/yr decrease in discharge. When including human water use, the simple hydrological budget of P = Q + E + TWS does not hold, as it did for the land cover experiments, where the land cover-induced change in Q was compensated by the change in E and terrestrial water storage TWS (as P did not change between the experiments). For human water use in PCR-GLOBWB there is an additional source of water besides precipitation, namely desalinized water, and water is also lost through consumption. The latter consists of water abstracted for e.g. domestic, industrial and livestock demand which is not returned to the hydrological cycle. Changes in the hydrological budget are thus described by dDesalinized = dQ + dE + dTWS + dConsumption. With dDesalinized = 1, dQ = -1185, dE = 846 (including evapotranspiration from irrigation), dTWS = -185 and dConsumption = 499 km$^3$/yr the balance is practically closed, see Table 4. Note that despite the fact that dQ is not fully balanced by dE for human water use, locally the effect on evapotranspiration through irrigation is higher than the effect of land cover change, especially in water limited arid regions (further described in Section 3.3). Note that despite the fact that dQ is not fully balanced by dE for human water use, locally the effect on evapotranspiration through irrigation is strong, especially in water limited arid regions (further described in Section 3.3).

The combined impact of land cover change and human water use (HUM2000 minus LC1850) would be a reduction in total evapotranspiration of 42 km$^3$/yr, or 0.1%, and a discharge increase of 284 km$^3$/yr, or 0.6%, see Table 4. The effect of land cover change, increasing discharge, largely cancel out the effect of human water use, decreasing discharge. These global averages however mask spatial variability, with discharge changes due to land cover change covering both in- and decreases, see Fig. 5 and Table 3. Evapotranspiration is most sensitive to land cover change in most regions (Fig. 6(c)), but globally averaged the effects cancel out due to the strong impact of irrigation on evapotranspiration.

## 3.3 Changes in global hydrology: subbasin analysis

To further specify how the impacts of land cover and human water use vary amongst different land cover transitions and different climate zones, we use the subbasins defined in Section 3.1.

Specific changes in discharge per subbasin are represented in Fig. 8, showing discharge in experiment LC2000 versus LC1850 or HUM2000 for the different climate zones, with each color representing a land cover change. Land cover changes cause an increase in discharge in most subbasins, with most spread in the sign of change for the transition of short natural

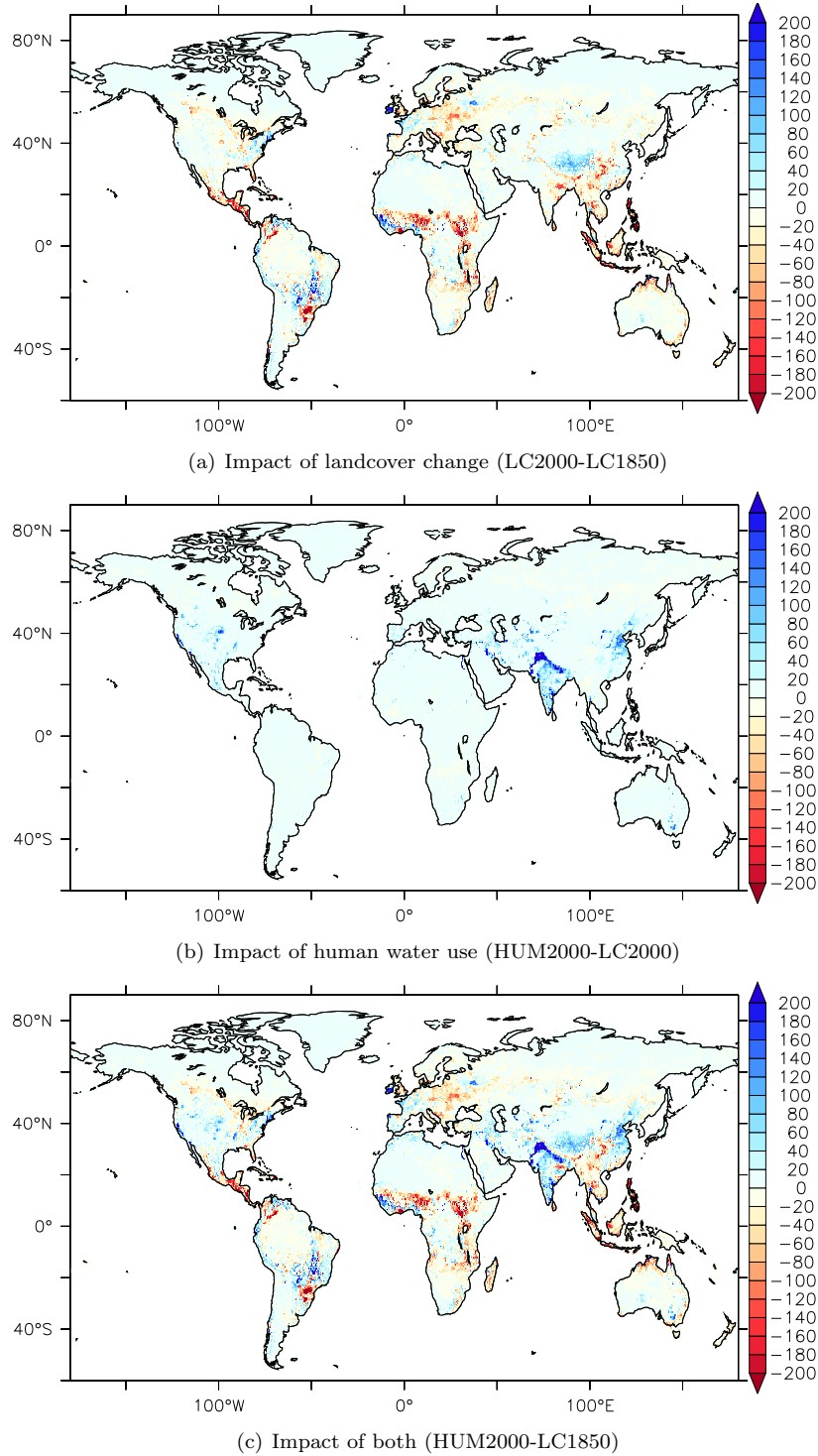

(a) Impact of landcover change (LC2000-LC1850)

(b) Impact of human water use (HUM2000-LC2000)

(c) Impact of both (HUM2000-LC1850)

**Figure 6.** Difference in annual total land surface evapotranspiration between the experiments, averaged over 1979-2010, in mm/yr.

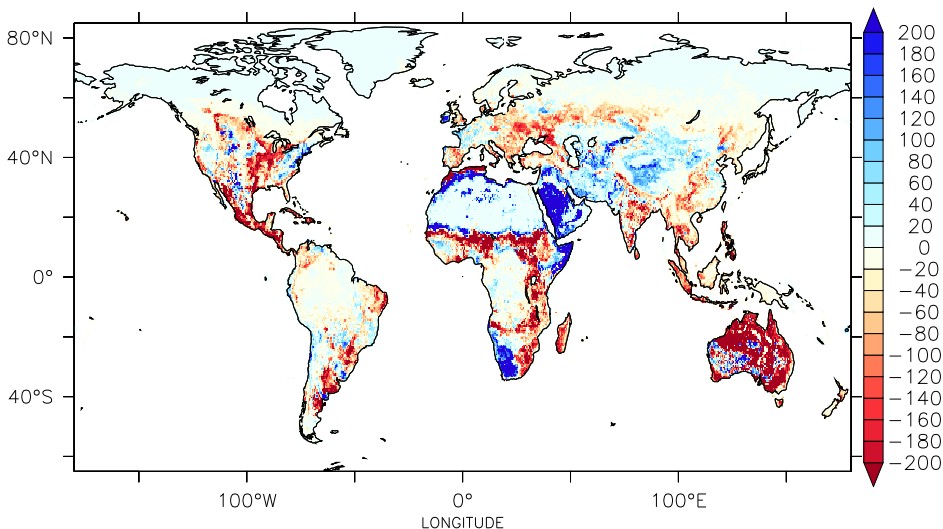

**Figure 7.** Difference in potential evapotranspiration, averaged over 1979-2010, between LC2000 and LC1850, in mm/yr (average of annual totals). Note that there is no difference in potential evapotranspiration between HUM2000 and LC2000.

to pasture. On average the largest increase occurs when natural vegetation is replaced with crops, followed by the transition from natural vegetation to pasture. Furthermore, the largest discharge changes occur in arid climates (B), especially when tall natural vegetation is replaced by crops. Areas where natural vegetation replaces crop or pasture ('other') generally experience a decrease in discharge (more accurately reflected in Fig. A6). Smallest discharge changes occur when short natural vegetation

is replaced by pasture, except in polar climates (E), where other transitions hardly occur (see Table 2). A similar picture arises when looking at relative changes in discharge (Fig. A6).

Changes in discharge per subbasin due to human water use are of the same order of magnitude overall, but have a larger effect in in warm temperate and snow climates (C, D) does human water use affect the discharge slightly more (Fig. 8). This could be related to population density and consequently high water demands in these areas. In all areas except polar climates

(E) land cover change increases discharge, while human water use decreases discharge. Note that discharge within a subbasin may be affected by changes upstream.

Changes in actET and sensitivity to potET per subbasin are shown in Fig. 9. Areas more sensitive to changes in potET will have a stronger change in actET relative to the change in potET. Based on all subbasins (top left panel) there is an average reduction in actET, due to reduced potET as a result of land cover change (circles). Only the transition of natural to crop or

pasture ('other') results in higher actET. The transition of short natural to pasture also results in higher actET on average, but there is a large spread in both the magnitude and sign of change. There is also quite some spread for subbasins where tall natural vegetation is replaced by pasture, because natural vegetation and pasture can represent a variety of vegetation types (see Section 2.2). Areas where crop replaces natural vegetation generally show a larger reduction in actET and are most sensitive; changes in actET are high relative to changes in potET. This corresponds to larger discharge changes in such areas (Fig. 8)

compared to areas where pasture replaces natural vegetation. Only in polar climates (E) is the effect of changing short natural vegetation to pasture largest, but other transitions hardly occur here (Table 2). Conversion from short natural to pasture in other climate zones shows the least sensitivity, as the largest changes in potET occur mostly in arid climates (B, lower left panel of Fig. 9). Furthermore, in some areas conversion from short natural to pasture does not change potET, such as north of the Caspian Sea (compare Fig. 4 and 7). Despite the low sensitivity of actET to potET in arid climates (B), there is still a large reduction in actET when tall natural vegetation is replaced with crop or pasture, leading to a strong increase in discharge for these transitions in arid areas (Fig. 8). Sensitivity is highest in the wetter equatorial and warm temperate climates (A and C), in which there are more energy limited areas (potET $<$ P). Conversion of crop or pasture to natural vegetation ('other') results in higher evapotranspiration, with highest sensitivity in equatorial, warm temperate and snow climates (A, C and D).

Compared to land cover induced changes in actET, changes due to human water use are larger in arid and warm temperate climates (B and C). In arid areas, increased human water use in the form of irrigation has a strong effect on evapotranspiration as these areas are water limited areas (potET $>$ P), whereas the water-limitation means that these regions have a low sensitivity to changes in land cover. Warm temperate regions are less water limited but highly populated, which may explain the strong human impacts on actET. Note that there is no change in potET between HUM2000 and LC2000.

## 3.4 Changes in global hydrology: Budyko analysis

Another way of comparing the effects of land cover change to those of human water use is by representing river basins in the Budyko framework. Fig. 10 shows that human water use (HUM2000 vs LC2000) can strongly increase actET without changes in potET, moving basins towards or even over the supply limit of actET=P, by adding water to irrigated fields. The effect of human water use is larger than that of land cover in 33 out of the 100 basins plotted in Fig. 10, mostly in water-limited areas (potET $>$ P) where actET is not sensitive to the land cover-induced change in potET (see Fig.9) but where irrigation can greatly increase actET. Land cover changes affect both actET and potET, generally reducing both, except some areas, mainly water-limited basins where short natural vegetation is replaced by pasture. Such areas become more water-limited, while the majority of basins becomes more energy-limited (or less water-limited) due to land cover change.

## 4   Discussion

In this study we have shown that the impact of land cover change can be as important as the impact of human water use through e.g. irrigation, abstraction and dams. The latter reduces discharge through increased evapotranspiration over irrigated areas and reservoirs as well as water consumption, while land cover change effects vary spatially but overall reduce evapotranspiration and increase discharge. Conversion to crops leads to the largest reduction in evapotranspiration and hence increase in discharge, despite conversion to pasture covering a larger area. Areas converted to pasture may experience less evapotranspiration changes due to less change in vegetation types and therefore smaller changes in potential evapotranspiration, as well as the fact that a large part of this area is in arid, water-limited climatic conditions.

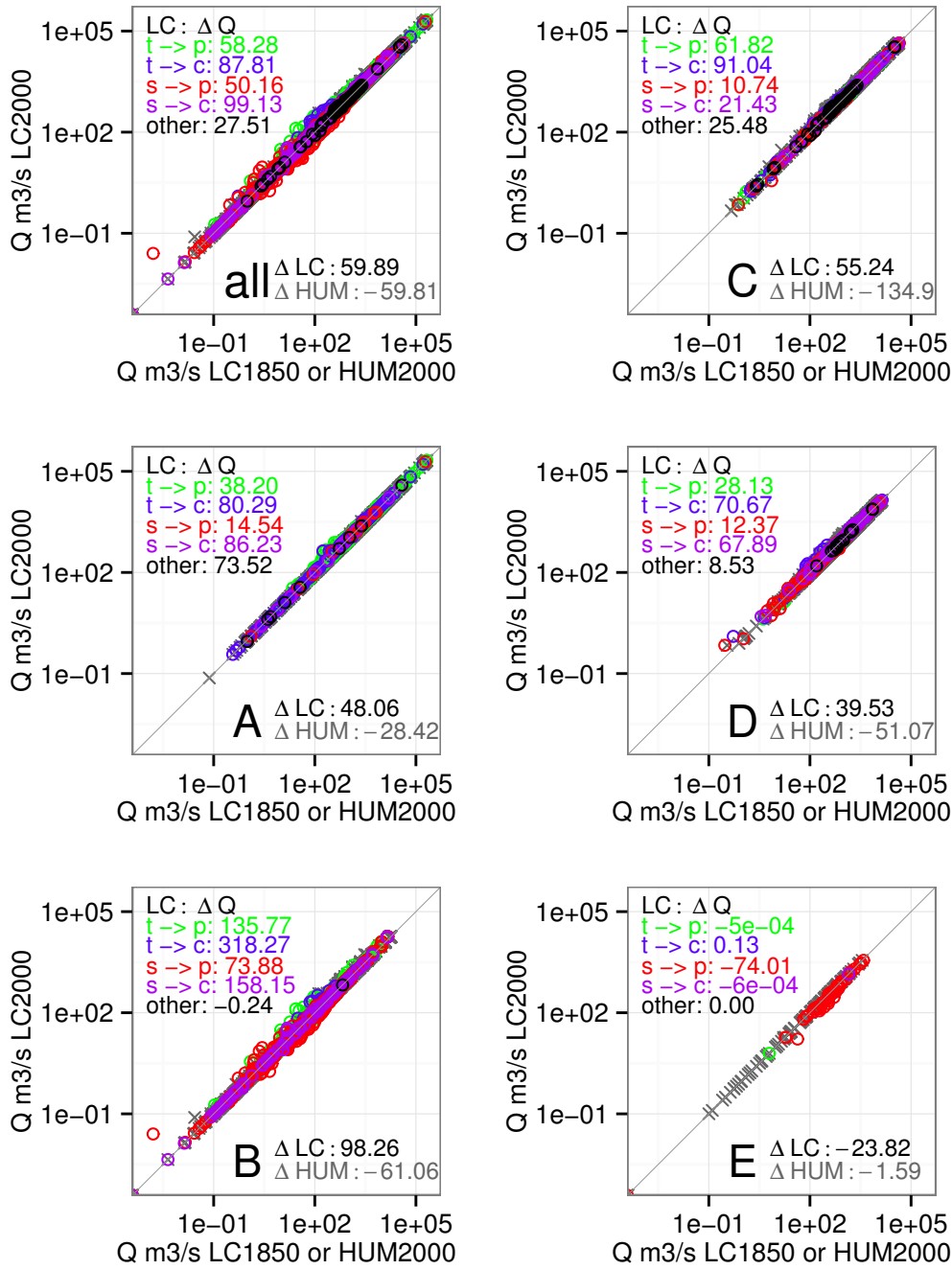

**Figure 8.** River discharge (Q) changes in m³/s per subbasin for all Köppen classes in the top left as well as per Köppen class. The axes are on a log-scale. Each circle color represents a land cover change: tall to pasture (green), tall to crop (blue), short to pasture (red), short to crop (purple), or other (black, crop or pasture to short or tall natural). No circles are drawn in subbasins where no land cover change occurs. Grey crosses represent discharge in LC2000 and HUM2000. Köppen class A is equatorial, B is arid, C is warm temperate, D is snow and E is polar climates. In each figure the top left numbers are the average discharge change per land cover change m³/s, in the bottom right are the total land cover changes as well as the changes due to human water use. Areas and number of subbasins per land cover change are given in Table 2.

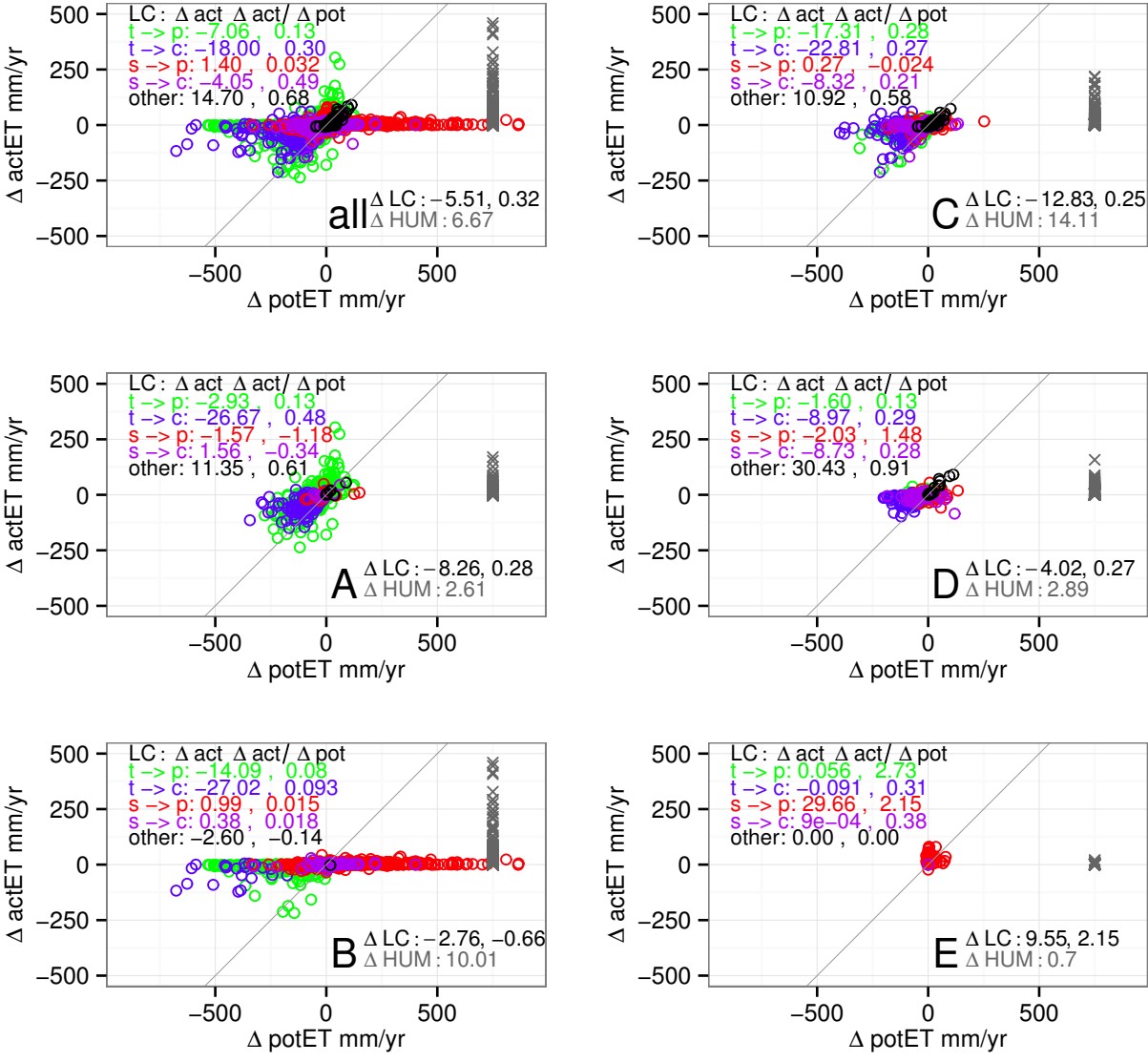

**Figure 9.** Changes in actual evapotranspiration (actET, y-axis) and potential evapotranspiration (potET, x-axis) from land per subbasin. Circles represent land cover change (LC2000 - LC1850), grey crosses represent human water use (HUM200 - LC2000). Note that there is no change in potET between HUM2000 and LC2000; the grey crosses have been moved along the x-axis for visiblity. Each circle color represents a land cover change: tall to pasture (green), tall to crop (blue), short to pasture (red), short to crop (purple), or other (black, crop or pasture to short or tall natural). The top left panel represents all subbasins, the other figures represent a Köppen class. A is equatorial, B is arid, C is warm temperate, D is snow and E is polar climates. In each panel the top left numbers are the average actET change per land cover change and the change in actET divided by the change in potET, in the bottom right are the total land cover changes (ΔLC, LC2000 - LC1850) as well as the changes due to redistribution (ΔHUM, HUM2000 - LC2000). No circles are drawn for subbasins where no land cover change occurs. In all figures the 1:1 line is drawn in grey. Note that in some cases the change in actET is larger than the change in potET, or of opposite sign. This generally occurs where changes in potET are small, such as high latitudes or the Amazon or Congo basins. It may also reflect areas where changes in e.g. soil moisture content or rooting depth alters the response to changed potET. Areas and number of subbasins per land cover change are given in Table 2.

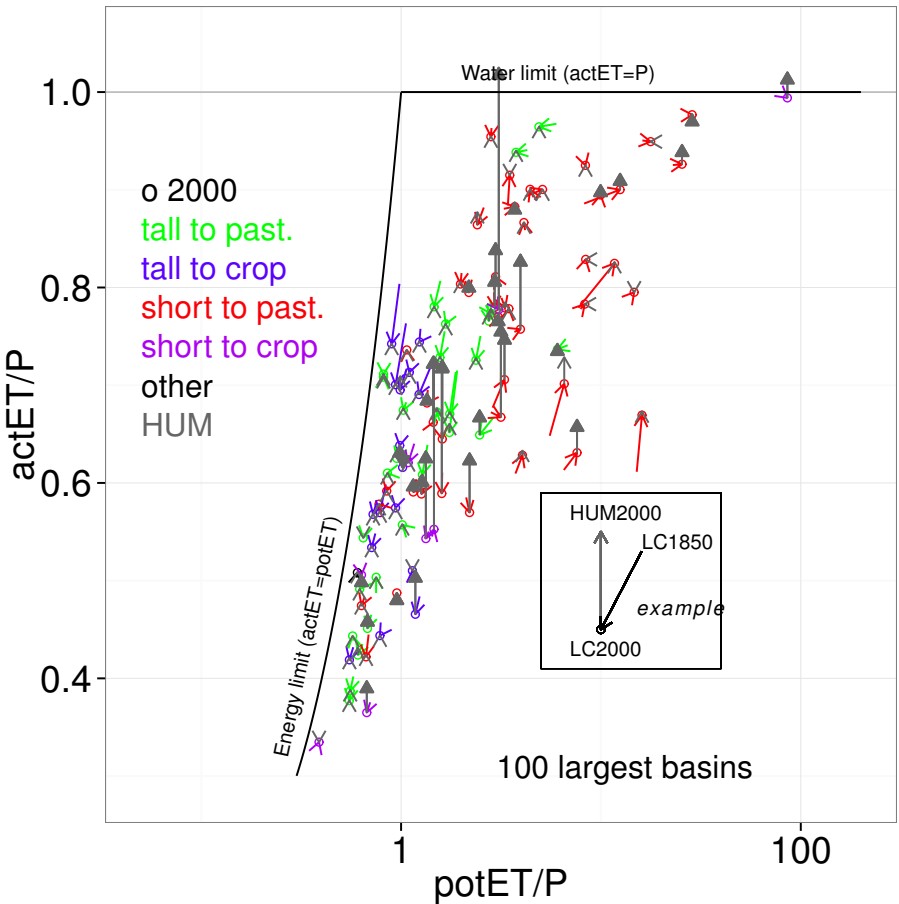

**Figure 10.** Annual climatological means of actET/P (y-axis) and potET/P (x-axis) for the 100 largest river basins (on our model grid). The x-axis is on a log scale. Circles represent actET/P and potET/P for the LC2000 experiment, colors indicate the land cover change, grey is for human water use. Arrows point from values for the LC1850 to those for the LC2000 experiment (colors), or from LC2000 to HUM2000 (grey), see the example. Solid arrowheads indicate that the change in actET/P induced by human water use is larger than the change induced by land cover. This occurs in 33 of the 100 basins. Here we re-classified the land cover changes for the entire basins, we did not group the subbasins that were used in Section 3.3. Figure A7 shows which basins were used for this Budyko analysis.

In this section we compare our results to previous studies on the impact of land cover change and / or human water use (Section 4.1), as well as provide a discussion on uncertainty due to input data (Section 4.2) and feedbacks that are not included (Section 4.3). We acknowledge that results are not only sensitive to input data but also to model physics, resolution and parameterization. A detailed discussion on model uncertainty is left out as our experiments are set up as sensitivity experiments; judging model performance compared to observations was not our goal.

## 4.1 Comparison to previous studies

Our results are generally in line with previous studies, stating that land cover change reduces evapotranspiration and increases discharge, with land cover impacts of similar magnitude as the impact of human water use. Differences in magnitudes and patterns of changes may be explained by using different computational tools and models and different input data (see also Section 4.2). A brief overview of global studies of the impacts of land cover and / or human water use is shown in Table 5, more detail is given in the text here.

Gordon et al. (2005) report a reduction in evapotranspiration due to deforestation of 3000 $km^3$/yr and an increase due to irrigation of 2600 $km^3$/yr, comparing potential (natural) to actual (present-day) vegetation. Here we report an 888 $km^3$/yr decrease due to land cover change and 846 $km^3$/yr increase due irrigation and reservoirs. Our changes are smaller despite a larger area of change; Gordon et al. (2005) compare a fully potential (natural) vegetation to actual vegetation with a total area of change of $15.9x10^6$ $km^2$, with crop and grazing land replacing forest and woodland, while we find a reduction of natural vegetation (both tall and short) of $34.8x10^6$ $km^2$, replaced by crop and pasture, from 1850 to 2000. The reduction of tall natural vegetation alone is $17.3x10^6$ $km^2$ in our study. Gordon et al. (2005) only include deforestation, replaced by cropland or grazing land. The transition of tall natural vegetation to crop is causing the strongest decrease in evapotranspiration (actET) and increase in discharge in our study, followed by the transition of tall to pasture, but here it is balanced by a weaker response of the transition of short natural vegetation to crop or pasture and sometimes even an opposite response (such as conversion of crop back to natural vegetation or short natural vegetation to pasture in arid or polar climates). Results may also differ because Gordon et al. (2005) works with vegetation-specific coefficients, with for instance crop coefficients for a range of tall natural vegetation types but all grazing land (pasture) having the same values as natural grassland. This could explain a larger sensitivity of transition to grazing lands than the transition to pasture in our study, as pasture has spatially varying parameter values (like all land cover types) which in some areas are close to those of the natural vegetation it replaces in our study.

Rost et al. (2008b) have also addressed how global terrestrial evapotranspiration and discharge are impacted by land use and irrigation, using the dynamic global vegetation model LPJmL. Like Gordon et al. (2005) they use a 'potential' natural vegetation, whereas we use the 1850 land cover to compare to present-day (in our case 2000) land cover. Their impact of land cover change on actET (-2361 $km^3$/yr, -3.8%) and discharge (2349 $km^3$/yr, 6.6%) is larger than the changes we find here (-1.6% and 1.9% respectively). Water redistribution includes only irrigation in Rost et al. (2008b), so they find smaller human water use induced changes than our study where we also include dams, water abstraction and consumption. Using only renewable water sources for irrigation they find increased actET of 483 $km^3$/yr (0.8%) and reduced discharge of 579 $km^3$/yr (-1.5%). In our study, actET from the surface (excluding evaporation from water bodies) is increased by 701 $km^3$/yr (HUM2000 vs LC2000),

which is larger than the increased actET of Rost et al. (2008b) due to irrigation, despite their comparison to potential natural vegetation vs our comparison to 1850 conditions. This could be related to their finding of 483 km$^3$/yr being based on only renewable water sources. In PCR-GLOBWB fossil groundwater is included, but limits on abstraction of this nonsustainable source are enforced (see Methods). In the study of Rost et al. (2008b), including non-renewable water resources to ensure no water stress on irrigated crops increases the impact of irrigation on actET to 1325 km$^3$/yr.

Another study reaching similar conclusions to ours, despite using different methods and land cover parameterization, is Sterling et al. (2013). They investigated the impact of global land cover change on the terrestrial water cycle using observations as well as land surface modelling (ORCHIDEE). They find that land cover change can have a similar or greater impact than other major drivers (mainly climate change and water consumption and withdrawals). Furthermore, both our study as well as Sterling et al. (2013) find that conversion to crops causes the largest volume change in evapotranspiration, despite conversion to pasture covering a larger area. The latter may be related to a large part of conversion to pasture occurs in arid regions which are least sensitive to ET changes. The reduction in total evapotranspiration in our study (888 km$^3$/yr, 1.8%) is smaller than in theirs (3500 km$^3$/yr, 5%), which could be related to the larger anthropogenically impacted part of the global surface area in their 'present day' land cover (41%). This land cover is compared to a fully natural ('potential') land cover. Here, we compare land cover of 2000, with 36% of the surface covered with crops or pasture, to that of 1850, with 10% anthropogenic land surface. Hence we essentially increase the anthropogenically impacted surface area by 26%. Furthermore, we note that Sterling et al. (2013) include evaporation from reservoirs and wetlands in their study, with wetland loss causing strong reduction in evapotranspiration,while we neglect reservoirs in the LC2000 and LC1850 experiments and wetlands are not included in any of our experiments. With a smaller change in evapotranspiration we also find a smaller increase in discharge (1.9% vs the 7.6% increase reported by Sterling et al. (2013)).

On a smaller scale, Haddeland et al. (2007) find increased runoff due to land cover change over North America and Asia using the Variable Infiltration Capacity model. They furthermore find that dams and reservoirs have the most important effects on river runoff, because reservoir operations can strongly change a river's hydrograph. Here we have not included seasonal changes, but acknowledge that indeed the effects can vary seasonally (Haddeland et al., 2006). The impact of changing land cover of 1900 to 1992 is similar to the impact of irrigation and reservoirs in Asia, while in North America the impact of irrigation and reservoirs is larger (Haddeland et al. (2007), their Fig. 6). Here we also find that at least three of the major North American rivers included in Table 3 are impacted more by human water use (Mississippi, Columbia, Colorado). In Asia, the Mekong river is impacted more by land cover change. Human water use has a larger impact on especially the Indus, but also the Ganges-Brahmaputra, Yangtze, Yenisey, Ob, Lena and Yellow rivers. Furthermore, in Haddeland et al. (2006) the consumptive irrigation water use is estimated at 98 km$^3$/yr for North America and 509 km$^3$/yr for Asia, which is on the same order of magnitude as the 776 km$^3$/yr of water lost globally through evaporation over irrigated areas stated in our HUM2000 experiment. Biemans et al. (2011) report a global reduction 930 km$^3$/yr (2.1%) in discharge due to irrigation over the 20th century using the LPJmL model, close to the 1185 km$^3$/yr (2.5%) reported in our study. Irrigation water supply from reservoirs is 460 km$^3$/yr in their study, versus 776 km$^3$/yr evaporation from irrigation in our HUM2000 experiment from reservoirs as well as other sources (precipitation, rivers, groundwater).

The reduction in evapotranspiration due to land cover change in our study is closer (globally averaged) to the 1260 km$^3$/yr (diagnosed based on ET products) or 760 km$^3$/yr (simulated, LUCID LSMs) reported by Boisier et al. (2014). They compare 1992 to 1870 instead of a fully (potential) natural vegetation as in the studies above. However, note that Boisier et al. (2014) report large uncertainty margins on these numbers (1260±850 and 760±720 km$^3$/yr, see Section 4.2). Sterling et al. (2013) also report a large range of estimates (their Figure 2). This implies that the actual values are rather uncertain, as also exemplified by the various numbers reported above, but all studies point to decreased evapotranspiration due to land cover change. Thus, despite the idealized set-up or our experiments (for instance keeping land cover and water use fixed at values for the year 2000 in HUM2000), the range of values previously reported, and the values reported here being smaller than the model bias (see Section 2.4), the findings for the impacts of land cover and water use are in line with those previously reported.

## 4.2 Uncertainty in input data

The numbers presented in this study are dependent on not only the model used but also the input data. Here we use fractions of crop and pasture from the harmonized land use data of Hurtt et al. (2011), which shows some differences to the SAGE dataset of Ramankutty and Foley (1999), used by e.g. Gordon et al. (2005); Haddeland et al. (2007); Sterling et al. (2013). Haddeland et al. (2007) discuss some differences between SAGE, the dataset of Ramankutty and Foley (1999), and HYDE (Klein Goldewijk et al., 2011), which is used for the historical part of the dataset of Hurtt et al. (2011). For present-day, SAGE has 15% of global land area identified as cropland, while HYDE identifies 11% as cropland and 23% as pasture. Furthermore, deforestation in SAGE is 11.5% but 17% in HYDE. Hence using different sources of crop and pasture cover, combined with each study / model representing vegetation parameters in their own way, introduces differences in results.

Even in studies aimed at representing present-day hydrological conditions, a different land cover dataset can impact the results. Müller Schmied et al. (2014) present a sensitivity analysis of the global hydrological model WaterGAP to input data, model structure, human water use and calibration. They find that using different land cover products (MODIS vs GLCC) has a bigger effect on grid cell fluxes such as actET and Q, than human water use. At the global scale this effect averages out and human water use is more important for global sums of Q, while land cover is more important for actET. Our study agrees on the latter, but here we find that land cover change has a comparable effect on global discharge sums as the effect of human water use, which may be related to the fact that we apply a larger land cover change (1850 vs 2000 instead of two different land cover datasets for present-day). However, both studies underline the importance of land cover in terrestrial hydrological fluxes.

Boisier et al. (2014) discuss land-use induced changes in actET based on various observations as well as model studies, reporting a decrease in actET of 1260±850 and 760±720 km$^3$/yr respectively, based on LUCID intermodel comparison of land surface models. Differences can arise from distinct land surface parameterizations in models as well as different land cover maps and different crop evapotranspiration rates in different land cover products. Therefore, Boisier et al. (2014) state that 'comparisons between independent estimates might be misleading', as one needs to take into account different computational methods or models, different land cover input products, as well as wether or not a study includes e.g. irrigation.

Concerning the impact of human water use, there is some spread in literature in the actual estimates as well. Part of this spread results from taking into account different aspects of human water use, whether it be only irrigation (e.g. Rost et al., 2008b) or also reservoirs (e.g. Haddeland et al., 2007; Sterling et al., 2013). Here we take both into account, but keep irrigated areas and reservoirs fixed at 2000, in order to set up sensitivity experiments in line with the land cover experiments in which land cover is kept fixed during the experiment. One potentially influential assumption we make is that the relative cover of rainfed and irrigated crops is fixed according to the MIRCA dataset (Portmann et al., 2010). In our HUM2000 experiment, irrigation can be applied over an area of $2.99 \times 10^6$ km$^2$ (paddy and non-paddy combined), close to the $3.07 \times 10^6$ km$^2$ equipped for irrigation according to FAO (Siebert et al., 2013). However, the distribution of irrigated areas is different, here we for instance do not include irrigated areas west of the Black Sea, which are included in FAO based irrigated areas as used by e.g. Wada et al. (2014). Taking a different pattern of irrigated areas, or reservoirs and human water demand from another year than 2000, would likely influence the reported changes in actET and discharge in HUM2000 compared to LC2000. Lastly, we overestimate the irrigated area in 1850 by applying fixed rainfed and irrigated crop cover ratios from MIRCA, but this should not affect our land cover induced changes because in the LC1850 experiment no irrigation is applied, all crops are rainfed.

Despite the variety in estimates of land cover and / or human water use impacts in literature, the general conclusion that land cover changes reduce actET and increase discharge, with a similar order of magnitude as the impact of human water use, is robust amongst studies.

### 4.3 Uncertainty due to feedbacks not included

Our aim was to use idealized sensitivity experiments to investigate the direct effects of land cover effects as well as the effects of human water use. The actual values may be affected by e.g. model physics and parameter values or different input sources as discussed above. Also, feedbacks that are generally not included in global hydrological model studies may affect the outcomes. In this paper we use the same climatic forcing for all experiments, with no feedbacks to the atmosphere. We therefore do not include the effect that changing evaporation has on precipitation, which is known to affect precipitation particularly over irrigated areas (e.g. Tuinenburg et al., 2014; Cook et al., 2015; Pei et al., 2016). We note that by using reanalysis data as climatic forcing (CRU-ERA-Interim, see Section 2.1) the observed changes in precipitation that reflect such feedbacks are likely included.

Furthermore, by applying climatic forcing representative of present-day to the LC1850 experiment we do not take into account that besides a change in land cover, the climatic forcing around 1850 was slightly different. Neither do we take into account that the vegetation parameters used may be different under different climate conditions, such as the instance lower $CO_2$ levels and temperatures in 1850. However, by keeping the climate forcing and the parameter values per land cover type equal, we can investigate the direct effect of land cover change.

PCR-GLOBWB is a hydrological and water balance model, it does not compute the energy balance; the potential evapotranspiration can be computed from e.g. radiation and vapor pressure and then be provides as a boundary condition, but the model does not compute how the land surface affects e.g. the radiation fluxes back to the atmosphere. We thus cannot compare how including the land-atmospheric energy balance changes as a result of land cover change or human water use and how this

may affect the water balance. Land surface models such as ORCHIDEE used in (Sterling et al., 2013) do typically include the energy balance. For a full inclusion of both the energy balance and precipitation feedbacks, general circulation models are used, but those studies typically do not focus on the water balance, nor have an accurate representation of the interaction between the hydrological cycle and human water use.

## 5  Conclusions

In this study we used the PCR-GLOBWB global hydrological model to investigate the hydrological impacts of global land cover change as well as human water use. Land cover change is broken down into transitions of short or tall natural vegetation into crop or pasture, as well as a few areas where natural vegetation returns. Globally averaged, changing the land cover from 1850 to that of 2000 decreases evapotranspiration by 888 km$^3$/yr (1.5%), resulting in a discharge increase of 901 km$^3$/yr (1.9%). There is spatial variability in the response to land cover change, especially for the transition of short natural vegetation to pasture. The strongest responses generally occur when tall natural vegetation is replaced by crops and in energy-limited equatorial and warm temperate regions. The globally averaged response to the inclusion of human water use is a discharge decrease of 1185 km$^3$/yr, on the same order of magnitude as the impact of land cover change on discharge. Part of the discharge decrease is related to enhanced evapotranspiration over irrigation and reservoirs (846 km$^3$/yr), which can result in larger evapotranspiration changes than land cover change locally. The exact numbers reported here depend on choices in input data and model set-up, but we conclude that land cover change needs to be included in studies assessing the anthropogenic impact on the global hydrological cycle.

**Appendix A: Supplementary figures**

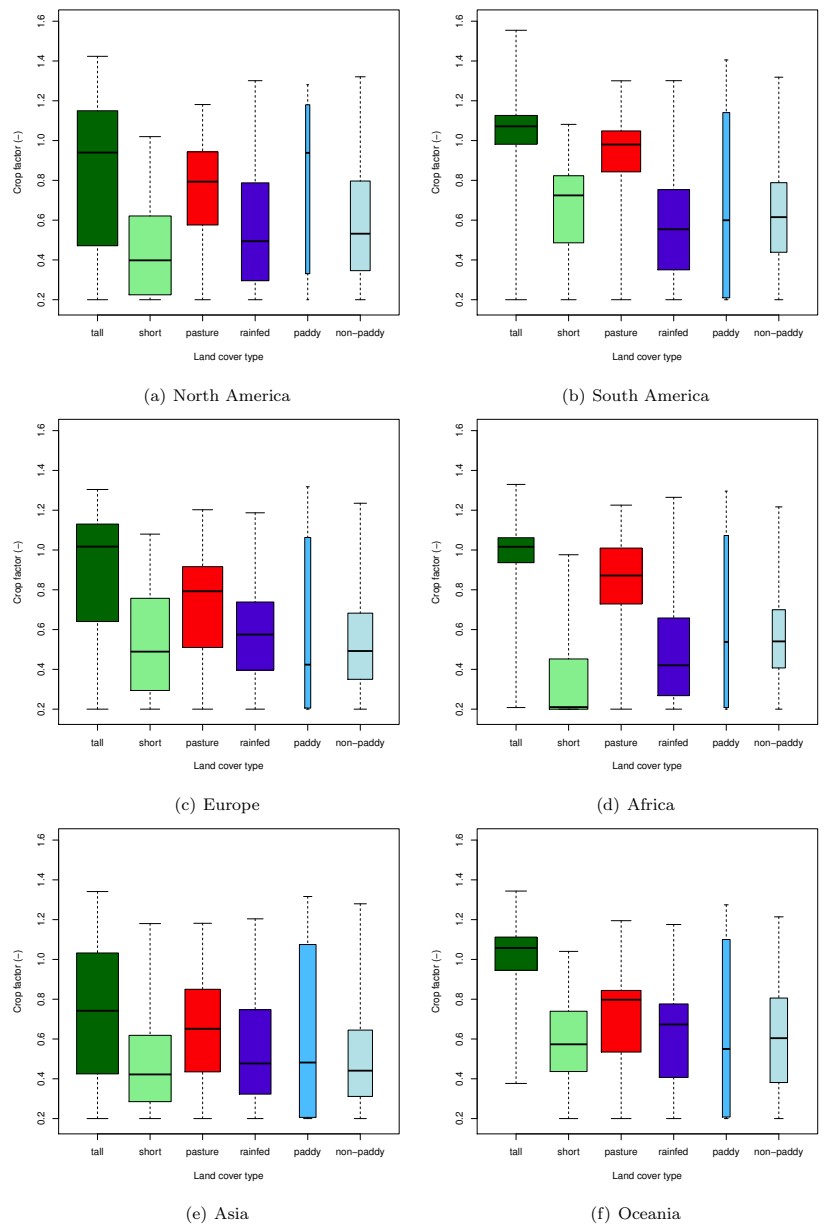

**Figure A1.** Variation in crop factor ($k_c$) in LC2000, used to compute land cover-specific potential evapotranspiration ($ET_{pot} = k_c * ET_{refpot}$), per continent and per land cover type. All daily $k_c$ values are included. Box plots indicate the minimum and maximum values by the whiskers, the interquartile range (between the first and third quartile) by the box and the median value by the black line within the box. Width of the boxes is proportional to the amount of grid cells within a continent where a land cover type is present. The spread for paddy irrigated crops is high because $k_c$ is high during the growing season but rather low (near 0.2) outside the growing season. Continental masks where derived using basins (see Fig. A2), with North and South America separated through central Mexico, Europe and Africa separated through the Arabian Peninsula, Europe and Asia separated through the Ural mountains, and Asia and Oceania separated roughly along the border of Malaysia and Indonesia.

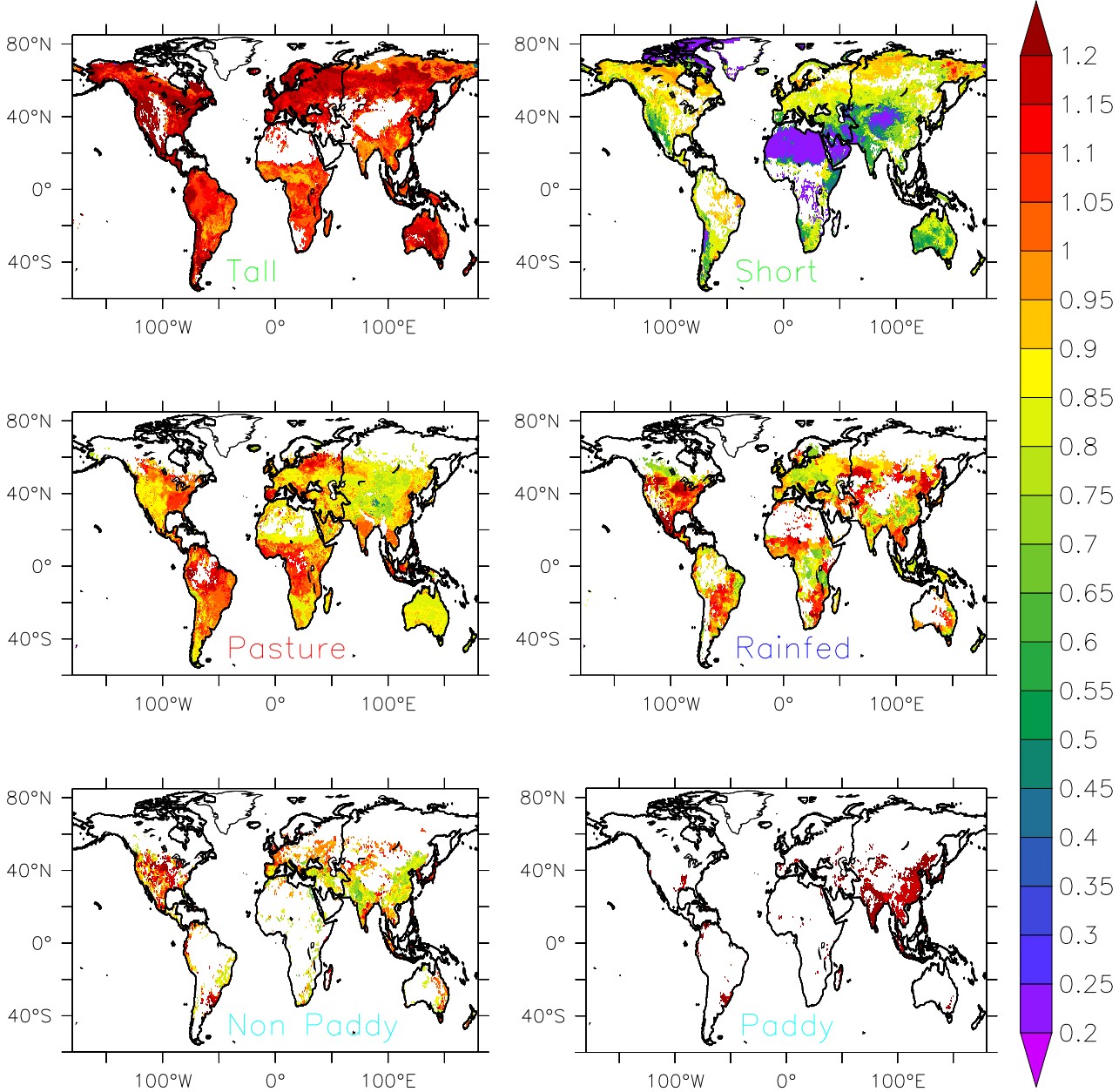

**Figure A2.** Maximum crop factors ($k_c$) in LC2000 per land cover type, used to compute land cover-specific potential evapotranspiration ($ET_{pot} = k_c * ET_{refpot}$). For each grid cell the maximum value is given, which may occur at different times during the year. Values are given where a land cover type covers more than 1 % of a grid cell. Black lines indicate the masks used for the continents in Figure A1. Note that short natural vegetation includes desert areas where the crop factor is set to a minimum value of 0.2, hence the low crop factors for short natural vegetation in e.g. Africa (see Fig. A1). Low crop factors for short natural vegetation in North Africa derive from Arctic vegetation.

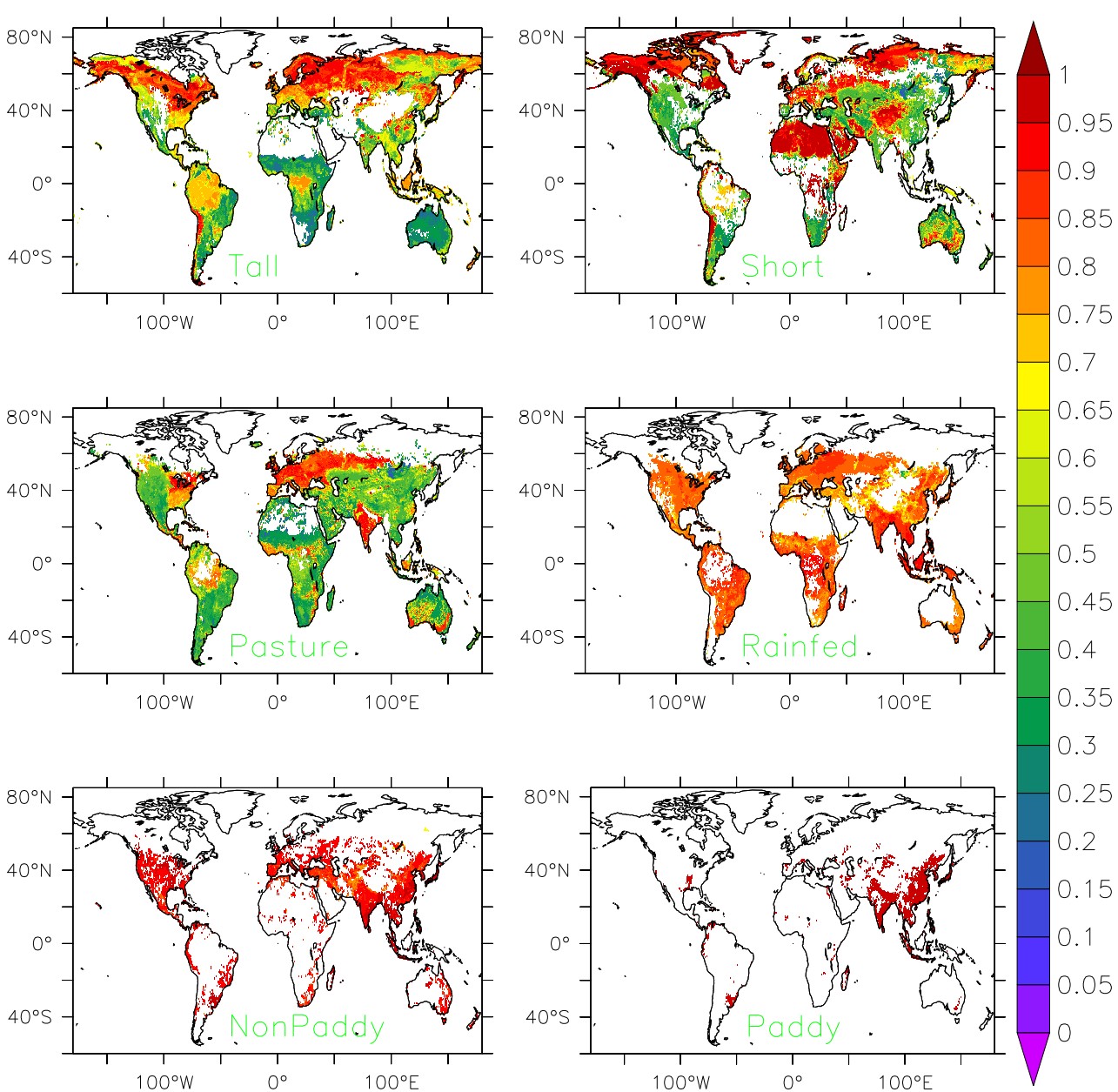

**Figure A3.** Root fraction in soil layer 1 (upper soil layer, reaching 0.13 to 0.3m depth) in LC2000 per land cover type, used to compute land cover-specific transpiration. A fraction of 1 indicates that all roots are in the upper layer, i.e. no water is taken by the roots from the deeper soil layer. Values are given where a land cover type covers more than 1 % of a grid cell.

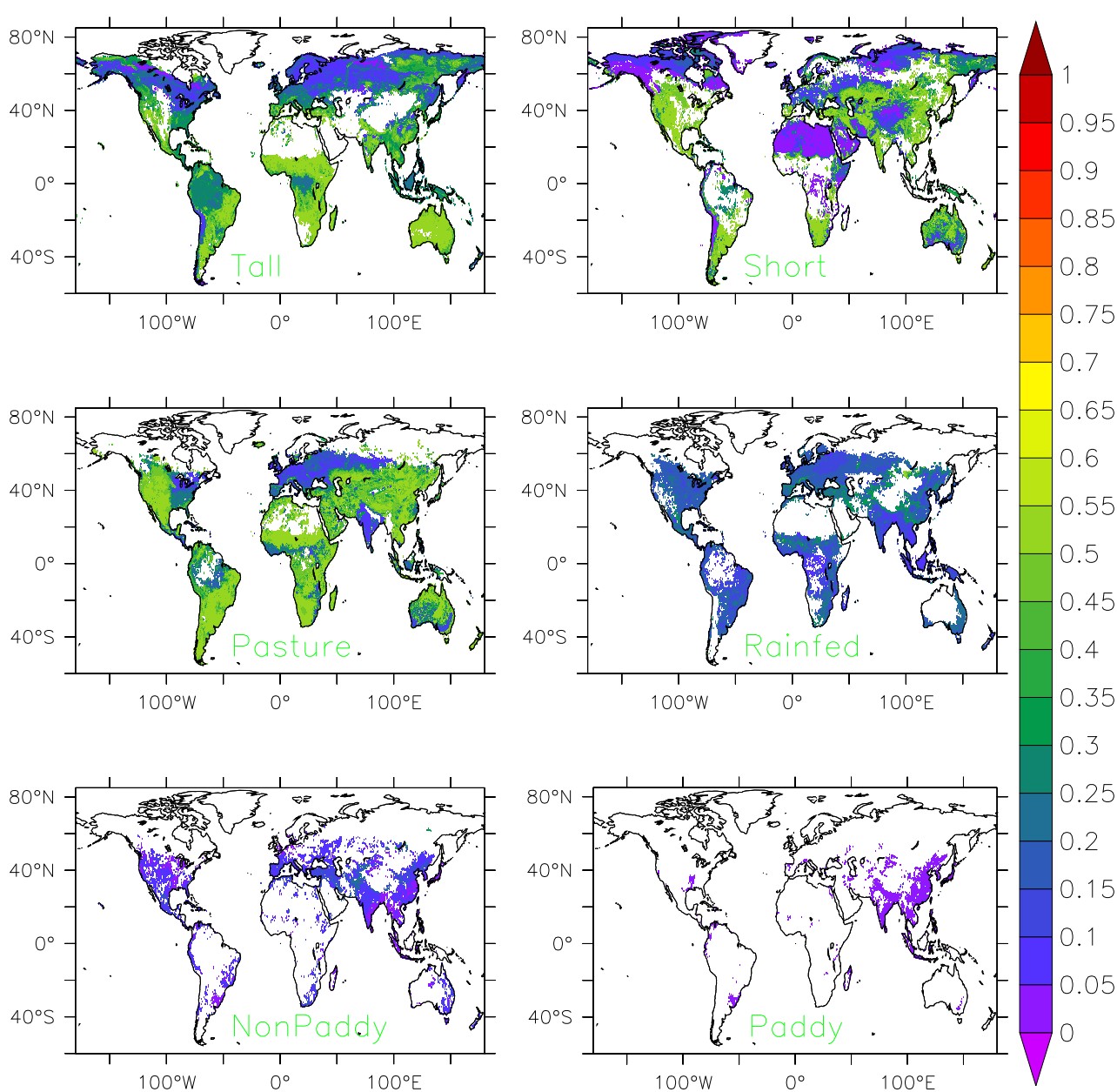

**Figure A4.** Root fraction in soil layer 2 (lower soil layer, reaching 0.52 to 1.2m depth) in LC2000 per land cover type, used to compute land cover-specific transpiration. The higher the root fraction, the more root is in the lower soil layer (and thus able to pick up moisture from both layers). Values are given where a land cover type covers more than 1 % of a grid cell.

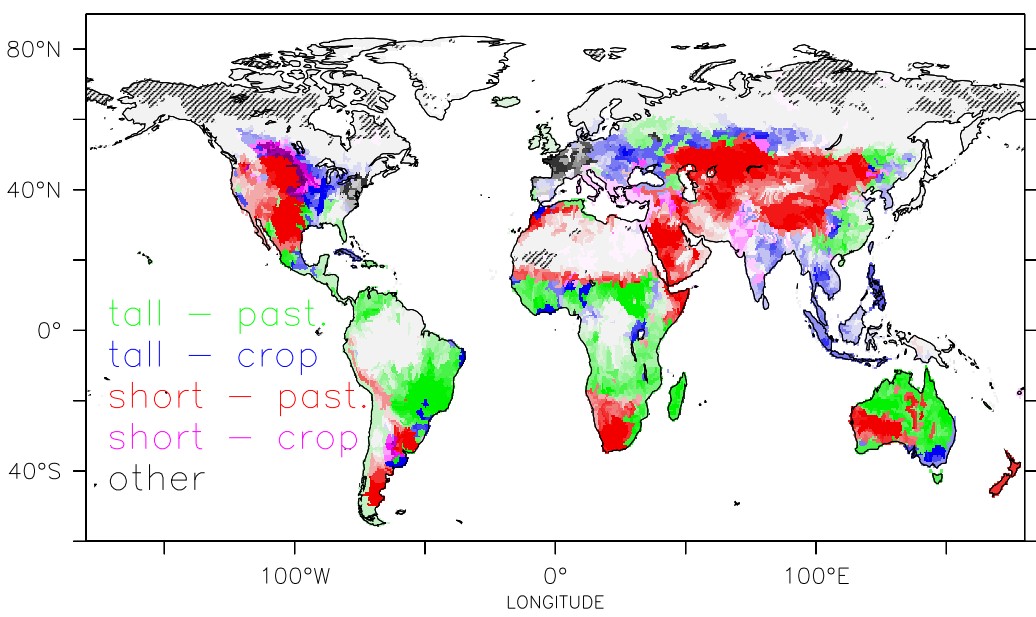

**Figure A5.** Areas covered by selected subbasins (3995 in total, see Table 2 and Section 3.1). Green areas indicate where the main change in the subbasin is from tall natural to pasture, blue represents tall natural to crops, red represents short natural to pasture and purple represents short natural to crops. Grey indicates subbasins where the main change is from crops or pasture to tall or short natural (e.g. in western Europe, eastern North America). Dashed black indicates where there is no land cover change (e.g. high polar latitudes). Color intensity indicates the change in natural vegetation, with near-white indicating almost no change and most saturated colors indicating that tall or short natural vegetation has de- or increased at least 50%. Table 2 shows the surface areas in each of these areas.

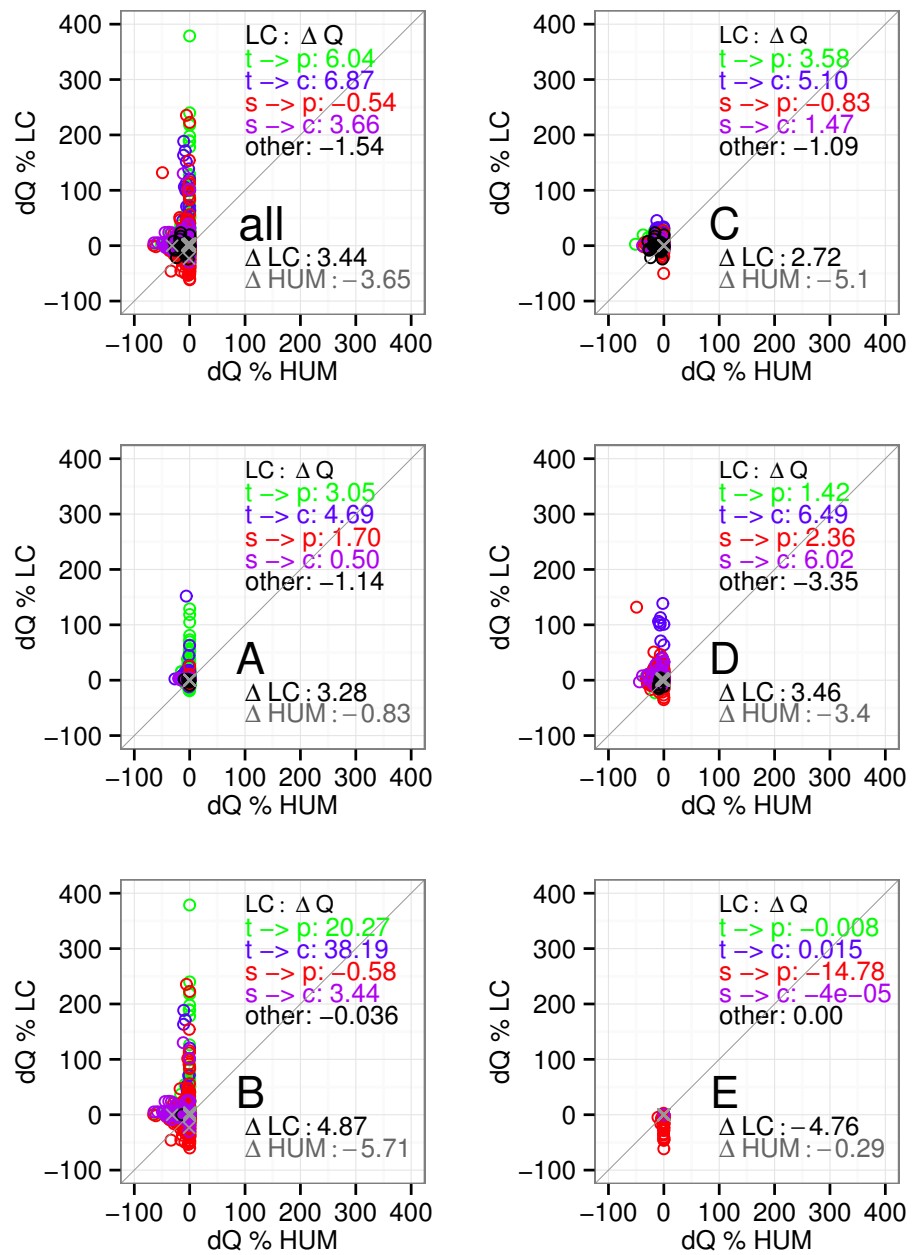

**Figure A6.** River discharge (Q) changes in % per subbasin for all Köppen classes in the top left as well as per Köppen class. Change due to human water use is represented on the x-axis (HUM, (HUM2000-LC2000)*100/LC2000), change due to land cover change is given on the y-axis (LC, (LC2000-LC1850)*100/LC1850). Each circle color represents a land cover change: tall to pasture (green), tall to crop (blue), short to pasture (red), short to crop (purple), or other (black, crop or pasture to short or tall natural). Grey crosses represent subbasins where no land cover change occurs. Köppen class A is equatorial, B is arid, C is warm temperate, D is snow and E is polar climates. In each figure the top right numbers are the average discharge change per land cover change in %, in the bottom right are the total land cover changes as well as the changes due to human water use in %. Areas and number of subbasins per land cover change are given in Table 2. One subbasin in B, short to pasture, was removed from this figure; with very low Q ($< 1$ m$^3$/s), dQ in this basin became $> 1000$%. Figure 8 shows the absolute changes.

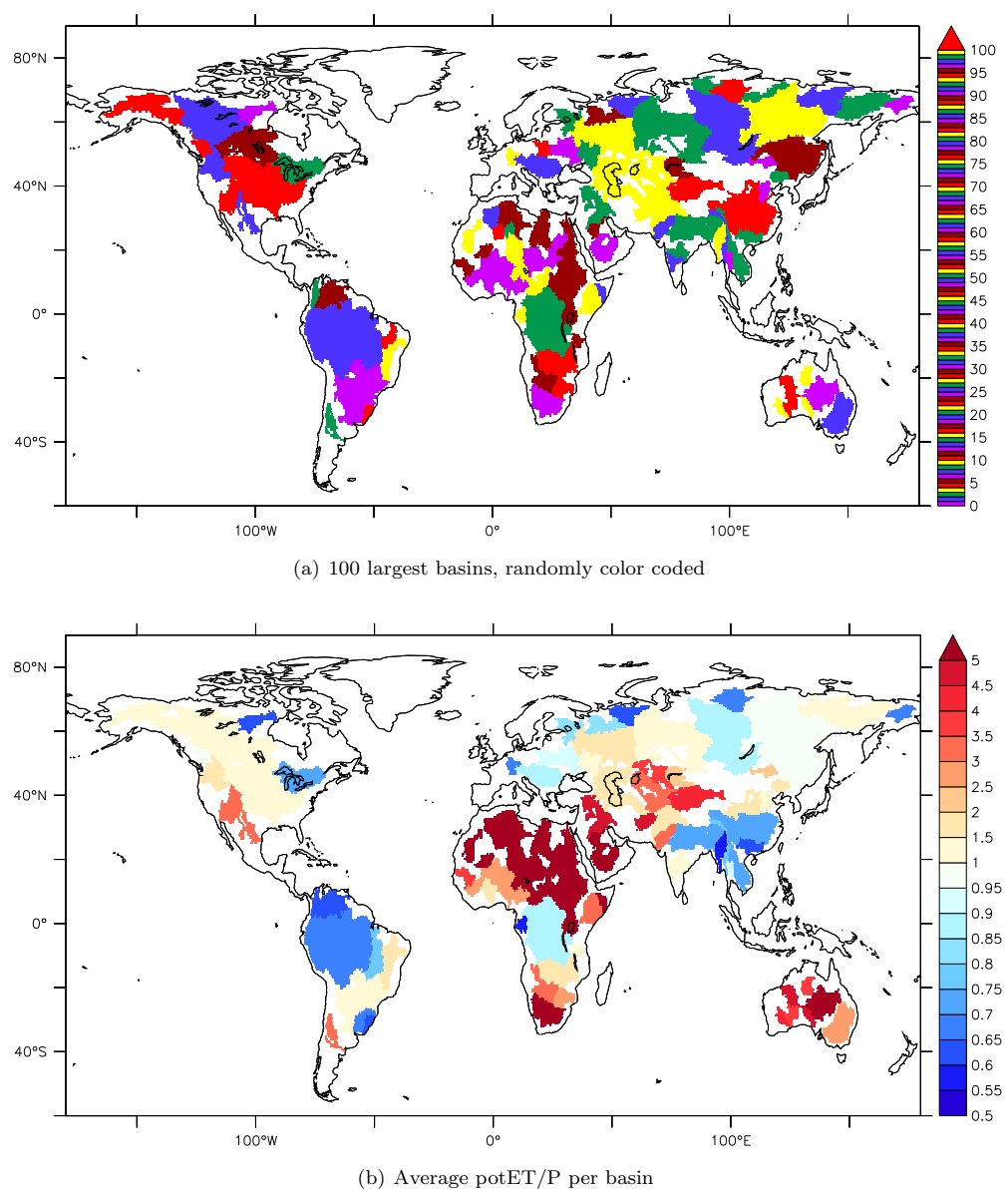

(a) 100 largest basins, randomly color coded

(b) Average potET/P per basin

**Figure A7.** The 100 largest basins on our model grid (a) and the average potET/P per basin (b). potET (potential evapotranspiration) is taken from experiment LC2000, annual averages of potET and P (precipitation) were used. Blue areas are energy limited (potET < P), red areas are water limited (potET > P), with darker colors indicating a stronger energy or water limit.

*Author contributions.* All authors contributed to the design of the experiments and the writing of this manuscript. J. Bosmans and R. van Beek prepared the land cover parameterization. E.H. Sutanudjaja and J. Bosmans adapted the model code of PCR-GLOBWB to run with the 6 land cover types used in this study.

5   *Competing interests.* The authors declare that they have no conflict of interest.

*Acknowledgements.* The authors would like to thank dr. Menno Straatsma, dr. Niko Wanders and dr. Yoshi Wada for valuable discussions and technical support while setting up these experiments and analyses. This study is funded by Utrecht University through its strategic theme Sustainability, sub-theme Water, Climate & Ecosystems.

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

| Experiment | Land cover | Water use |
|---|---|---|
| LC1850 | 1850 | No |
| LC2000 | 2000 | No |
| HUM2000 | 2000 | Yes |

**Table 1.** Overview of experiments. Water use includes water for domestic, industrial and livestock use, irrigation, dams and reservoirs as well as desalinized water used in coastal areas (Wada et al., 2014).

| | A | B | C | D | E | Total |
|---|---|---|---|---|---|---|
| Tall to pasture $10^6$ km$^2$ | 22.7 | 7.0 | 7.4 | 6.0 | 0.0 | 43.1 |
| *# subbasins* | *676* | *244* | *213* | *200* | *1* | *1334* |
| Tall to crops $10^6$ km$^2$ | 7.7 | 0.8 | 6.1 | 12.2 | 0.1 | 26.9 |
| *# subbasins* | *227* | *31* | *190* | *348* | *1* | *797* |
| Short to pasture $10^6$ km$^2$ | 0.6 | 28.4 | 3.4 | 8.2 | 5.4 | 45.9 |
| *# subbasins* | *18* | *820* | *99* | *240* | *47* | *1224* |
| Short to crops $10^6$ km$^2$ | 0.3 | 3.2 | 2.3 | 2.0 | 0.0 | 7.8 |
| *# subbasins* | *14* | *115* | *52* | *79* | *1* | *261* |
| Other $10^6$ km$^2$ | 0.0 | 0.1 | 1.6 | 1.1 | 0 | 2.9 |
| *# subbasins* | *9* | *1* | *50* | *15* | *0* | *75* |
| noLC $10^6$ km$^2$ | 0.0 | 0.5 | 0.0 | 3.7 | 2.6 | 6.8 |
| *# subbasins* | *17* | *28* | *10* | *153* | *96* | *304* |
| Total $10^6$ km$^2$ | 31.2 | 40.0 | 20.9 | 33.2 | 8.1 | 133.4 |
| *# subbasins* | *961* | *1239* | *614* | *1035* | *146* | *3995* |

**Table 2.** Area (in $10^6$ km$^2$ and number of subbasins) per land cover change and per Koppen-Geiger classification, based on 2000 minus 1850 land cover. A represents equatorial climates, B is arid, C is warm temperate, D is snow and E is polar (Kottek et al., 2006). Subbasins are divided into land cover change groups based on which natural land cover reduces most and which anthropogenic land cover increases most in a subbasin. Rainfed and irrigated crops are grouped together, as this subdivision will be used to analyse the impact of land cover change, where all crop land cover types are rainfed (LC2000 vs LC1850). 'Other' refers to those areas where tall or short natural vegetation is replacing crops or pasture. 'noLC' refers to subbasins where no land cover change occurs. See also Fig. A5.

| River | LC1850 | LC2000 | HUM2000 | ΔLC (%) | ΔHUM (%) |
|---|---|---|---|---|---|
| Amazone | 6642.5 | 6652.8 | 6648.9 | 10.3 (0.2) | -3.9 (0.1) |
| Orinoco | 1437.8 | 1454.5 | 1449.5 | 16.7 (1.2) | -5.0 (-0.3) |
| Uruguay | 314.9 | 327.7 | 324.5 | 12.7 (4.0) | -3.1 (-1.0) |
| MacKenzie | 172.1 | 174.5 | 172.8 | 2.4 (1.4) | -1.7 (-1.0) |
| Congo | 2116.4 | 2117.1 | 2116.7 | 0.7 (0.0) | -0.4 (0.0) |
| Nile | 439.0 | 549.9 | 502.3 | 110.8 (25.2) | -47.5 (-8.6) |
| Niger | 393.5 | 452.8 | 446.2 | 59.3 (15.1) | -6.6 (-1.5) |
| Dniepr | 70.4 | 89.9 | 76.5 | 19.5 (27.6) | -13.4 (-14.9) |
| Mekong | 537.7 | 555.5 | 548.2 | 17.9 (3.3) | -7.3 (-1.3) |
| Amur | 366.4 | 376.8 | 362.7 | 10.4 (2.8) | -14.1 (-3.7) |
| Ganges-Brahmaputra | 1211.0 | 1232.11 | 1182.1 | 21.1 (1.7) | -50.0 (-4.1) |
| Mississippi | 1060.9 | 1072.7 | 1022.9 | 11.8 (1.1) | -49.8 (-4.6) |
| Columbia | 163.6 | 165.1 | 152.9 | 1.6 (1.0) | -12.2 (-7.4) |
| Eufrat-Tigris | 77.3 | 78.5 | 50.6 | 1.2 (1.5) | -27.9 (-35.5) |
| Danube | 241.4 | 259.7 | 240.2 | 18.4 (7.6) | -19.6 (-7.5) |
| Yenisey | 437.4 | 442.9 | 435.8 | 5.6 (1.3) | -7.1 (1.6) |
| Ob | 361.4 | 372.1 | 359.3 | 10.7 (3.0) | -12.7 (-3.4) |
| Lena | 402.6 | 403.0 | 401.1 | 0.4 (0.1) | -1.9 (-0.5) |
| Yangtze | 1035.7 | 1063.1 | 1012.4 | 27.6 (2.7) | -50.8 (-4.8) |
| Murray-Darling | 169.5 | 176.9 | 164.2 | 7.3 (4.3) | -12.6 (-7.1) |
| Indus | 196.4 | 166.6 | 84.8 | -2.8 (-1.7) | -81.8 (-49.1) |
| Parana | 1410.4 | 1403.0 | 1381.1 | -7.4 (-0.5) | -21.9 (1.6) |
| Colorado | 37.9 | 36.5 | 23.9 | -1.3 (-3.5) | -12.6 (-34.5) |
| Orange | 33.1 | 32.5 | 28.2 | -0.6 (-1.8) | -4.2 (-13.1) |
| Yellow | 106.2 | 105.6 | 77.7 | -0.6 (-0.6) | -27.9 (-26.4) |
| Rhine | 79.2 | 78.4 | 71.3 | -0.8 (-1.0) | -7.1 (-9.1) |
| Global | 47010 | 47911 | 46726 | 901.2 (1.9) | -1185.3 (-2.5) |

**Table 3.** Discharge to the ocean from 26 rivers in km$^3$/yr for LC1850, LC2000 and HUM2000 in the first three columns, and differences (% given in brackets) in the last two columns. ΔLC represents land cover change (LC2000 minus LC1850), ΔHUM represents human water use (HUM2000 minus LC2000). Of these 26 river basins, the impact of land cover change is larger than that of human water use in the first 9. In the last 6 basins, both land cover as well as human water use act to decrease discharge. Note that discharge of the Nile is much larger than observed (pre-Aswan), PCR-GLOBWB does not perform well for the Nile so the absolute values need to be considered with caution.

|                | LC1850 | LC2000 | HUM2000 | dLC (%)     | dHUM (%)    | dTot (%)     |
|----------------|--------|--------|---------|-------------|-------------|--------------|
| Q              | 47010  | 47911  | 46726   | 901 (1.9)   | -1185 (-2.5)| -284 (-0.6)  |
| ET             | 58760  | 57872  | 58718   | -888 (-1.5) | 846 (1.5)   | -42 (-0.1)   |
| Desalinization | 0      | 0      | 1.2     | 0 (-)       | 1.2 (-)     | 1.2 (-)      |
| Consumption    | 0      | 0      | 499     | 0 (-)       | 499 (-)     | 499 (-)      |
| TWS            | 234    | 217    | 32      | -17 (-7.4)  | -185 (-85)  | -202 (-93)   |

**Table 4.** Overview of water balance terms in km$^3$/yr, with percentages in brackets for the last three columns except for desalinization and consumption as these are not included in LC1850 and LC2000. Q is the total global discharge. ET reflects total evapotranspiration. TWS is terrestrial water storage (including water bodies). Note that the positive values for TWS indicate a positive trend in each experiment in km$^3$/yr, reflecting a drift, and that TWS in HUM2000 is larger than in the other experiments (not evident from this table; in this case PCR-GLOBWB includes reservoirs and fossil groundwater).

| | dLC | dHum | Method | Notes |
|---|---|---|---|---|
| This study | ET -888 (1.5%) | ET +846 (1.5%) | PCR-GLOBWB | 6 land cover types, spatial variation representing various vegetation, |
| | Q + 901 (1.9%) | Q - 1185 (2.5%) | sensitivity exp. | crop and pasture types, fixed land cover and water use. |
| Gordon et al. (2005) | ET -3000 (4%) | ET +2600 | GIS | potential vs. actual vegetation, focus on deforestation |
| | | | | pasture represented by natural grasslands |
| Rost et al. (2008b) | ET -2361 (-3.9%) | ET +483 | LPJmL | potential vs. actual vegetation, using renewable water only for |
| | Q +2349 (+6.6%) | Q -579 | PFTs & CFTs | irrigation (dHum ET +1325 when non-renewable is included) |
| Sterling et al. (2013) | ET -3500 (5%) | - | GIS & | potential vs. actual vegetation |
| | Q +7.6% | - | ORCHIDEE | dLC includes wetland losses and reservoirs |
| Biemans et al. (2011) | - | Q -930 (2.1%) | LPJmL | Q decreased due to reservoir building and irrigation over 20th century |
| Boisier et al. (2014) | ET -1260 ± 850 | - | ET products | 1992 vs 1870 vegetation cover |
| | ET -760 ± 720 | - | LUCID LSMs | |

**Table 5.** Overview of studies assessing impacts of land cover change and / or human water use globally. Values given in km$^3$/yr. See Discussion in section 4.1.