# Peer review of "Hydrological impacts of global land cover change and human water use"

_Hydrology and Earth System Sciences, 2016_

## Referee Comment (RC1) · Anonymous Referee #1 · 10 Jan 2017

The paper describes a global assessment of hydrological impacts of land cover change and human water use for the period 1850-2000 and fits therefore well to the scope of the journal. The manuscript is well written and interesting; the figures shown in the manuscript are of good quality. Quantifying the effects of land cover change and water use on the hydrological cycle for such a long period is challenging and previous estimates varied considerably, depending on input data, models and assumptions used. Therefore, more research is needed to reduce these uncertainties. However, I think that a major revision is required before the manuscript may be considered for publication in HESS. My major points of criticism are:

General comments: 1) The authors quantify and compare the effects of land cover change and water consumption on evapotranspiration and river discharge. However, they assume that the third term in the water balance, the precipitation term, is not

affected by the changes in land cover and water use (at least there is no attempt to analyze changes in precipitation). This is a strong assumption that needs at least some discussion, because the authors present here a spatial analysis. There is growing evidence in the literature that both, land cover change and water use, modify precipitation patterns over large regions (see for example Pei et al., 2016 on the effect of irrigation on summer precipitation in the US). When irrigation results in increased ET, increased ET results in increased precipitation, and increased precipitation results in increased runoff. Consequently, the net effect of irrigation on river discharge may be much smaller than the results suggested by the authors. So the key question is certainly where water use and land cover change are taking place and in which region this will cause changes in precipitation (within the same watershed, outside of it but in another watershed or over the sea outside terrestrial surface). Answering this question is only possible by coupling a hydrological model with an atmospheric circulation model. This might be out of scope of the present analysis but the consequences of ignoring feedback mechanisms by changed precipitation patterns requires at least discussion.

2) One basic result of the study is that the effect of human water use on actual evapotranspiration is smaller than the effect of land cover change (page 11, line 6). However, the increase of ET by irrigation estimated by the authors seems to be very low compared to other studies. According to the present study, global ET is increased by irrigation by 377 km3 yr-1 (page 11, line 7) while other studies reported a much larger increase in ET by irrigation (for example, > 1000 km3 yr-1 between 1900 and 2005 according to Kummu et al., 2016). Why is that? Assuming that the uncertainty in additional ET created by irrigation is that large: how would this uncertainty then affect the basic conclusions drawn by the authors?

3) The authors explicitly pointed out that an analysis and discussion of the uncertainties involved in their estimates was not focus of the present analysis (page 14, lines 30-32). Nevertheless, these uncertainties exist and should be discussed. It is complex enough to simulate changes in ET on cropland because data for irrigated/rainfed crops and

the distinction between paddy and upland crops are available for recent years only, in addition simulation of ET for the period outside the cropping season requires many assumptions. Even more complex and extremely difficult is it to estimate changes in ET caused by the use of ecosystems as pasture. There are many different types of pasture characterized by distinct species composition, different proportion of woody biomass and different stocking densities. There are very intensive types of pasture with properties very similar to cropped surfaces and extensive pasture systems that hardly differ from natural vegetation. It remains completely unclear how the authors reflected this complexity in the parameterization of their model to estimate realistic changes in ET caused by the conversion of natural vegetation to pasture. In addition, there are large uncertainties about the historical extent of pasture. Currently available data sets differ considerably in their estimates. The authors mention these uncertainties in section 4.2 but it remains unclear how much the basic results of the study are impacted by these uncertainties. How robust are the results of the study? More description and discussion is needed.

4) The text section is often difficult to read because it contains too many numbers and reads to technical (e.g. section 4.1; section 3.2). I recommend to report the general findings in the text section and detailed results in tables. It may also help to develop a figure presenting the main results of the study (changes in terrestrial ET and discharge by water use and land cover change at global scale).

Minor comments: Abstract: Please report more in detail how the present study adds to a better understanding of the impact of lands cover change and water use on terrestrial hydrology. What is reported in the second part of the abstract represents more the state of knowledge but not new findings and conclusions from the present analysis.

Page 2, lines 25-29: This sentence is hard to understand. Please simplify.

Page 3, lines 5-9: Please simplify. Not nice to have brackets in brackets . . . .

Page 3, lines 28-30: ERA-Interim and CRU data often differ considerably, in particular

for precipitation and number of wet days. Is this not a problem when combining these two products?

Page 3, line 33: More description is needed how the different land cover types were parameterized to account for different types of pasture vegetation and crops. For example, the rooting depth may vary considerably even within the 6 major land cover classes used by the authors.

Page 6, section 2.3: How were reservoirs treated in LC1850 and LC2000?

Page 11, lines 6-7: "as evapotranspiration is only increased over irrigated areas". => This is an assumption made by the authors, however, in reality ET has also changed considerably in rainfed crops due to land use modification.

References: Kummu, M., Guillaume, J.H.A., de Moel, H., Eisner, S., Florke, M., Porkka, M., Siebert, S., Veldkamp, T.I.E. and Ward, P.J., 2016. The world's road to water scarcity: shortage and stress in the 20th century and pathways towards sustainability. Scientific Reports, 6, 38495.

Pei, L.S., Moore, N., Zhong, S.Y., Kendall, A.D., Gao, Z.Q. and Hyndman, D.W., 2016. Effects of Irrigation on Summer Precipitation over the United States. Journal of Climate, 29, 3541-3558.

---

## Referee Comment (RC2) · Anonymous Referee #2 · 15 Feb 2017

**Review of "Hydrological Impacts of Global Land Cover Change and Human Water use" by Bosmans et al.**

Generally the paper is interesting and well written. But for the moment it does not contribute originally to the literature on the impact of land use on the continental water cycle. Indeed this topic has been studied with many land surface models. But these are all numerical experiments which trust blindly that the parameters for the various vegetation types (Which have been tuned for the current climate and vegetation distribution.) apply to the original vegetation which existed before the human started to change landscapes massively in the mid 19$^{th}$ century. The authors acknowledge only partly this fact in the discussion section of the paper.
Coming from the global hydrological models community, the authors have a trump they should use. In contrast to classical LSMs, PCR-GLOWB is designed to simulate today's water usage and thus should simulate quite realistic river discharges in current conditions. Thus, the simulation HUM2000 should be much more realistic than the simulations on which the other land use studies are based. In other words, I would expect this study to present the realism of this simulation to argue for the quality of his study and its added value. Furthermore the use of the deviation from observed discharges could serve as an estimate of uncertainty and qualify the global averages changes in actET and discharge presented in section 4.1.

Thus, and before proposing a list of minor comments, I would suggest a major revision of this paper so as to present hydrological arguments as to why we should trust your numbers more than those of the cited papers. Else this paper will be just more noise on a topic where for the moment we are just guessing some numbers and anybody can propose "alternative facts".

Below you will find some minor comments which will hopefully help improve the paper. These comments also illustrate the major changes I would deem necessary to raise the level of this paper above the previous studies on this topic.

• Page 1, Line 21 : It is not true that few studies focus on including land use. Most land surface models used in the CMIP5 simulations apply a land use scenario. It could be true that the work is not very visible in the literature. I would attribute that to the fact that this is only a set of guestimates as the vegetation parameters are highly tuned and cannot claim to have any generality.
• Page 2, Lines 1-15 : In your review you do not mention that the picture is further muddied by the fact that in parallel to the land & water use change climate and aerosol loadings have evolved. Thus potET has a significant trend through modification of incident long-wave and solar radiation, atmospheric turbulence, water vapour pressure deficit and amplitude of the diurnal cycle. I can understand that this is outside of the scope of your study but these caveats need to be mentioned in the introduction. The literature is plentiful on this topic !
• Page 3, lines 25-30 : Please state clearly that you assume that the potET estimated for the period 1979-2010 is valid in 1870. To me this casts a big shadow over all land use studies but intellectual honesty requires that this is stated as a working hypothesis !
• Pages 18, line 18 : Can this impact on the Nile really be trusted ? The observed discharge in Aswan for the period 1871-1900 (i.e. before the first dam) is 112km3/y or about 3500 m³/s (What does PCR-GLOWB say ?) . Your combined change (HUM2000-LC1850) seems to be above 100m³/s, thus the amount of water in the Nile at Aswan should have increased !Observations indicate that the inflow into the great dam has not changed significantly since the end of the 19$^{th}$ century. On the other hand the amount arriving at the sea has dramatically been reduced. As you see, the value of your hydrological model is that you can check the reality of the predicted changes with the observations which date back to the 19$^{th}$ century. Based on my own experience the land use change proposed by LUH for the upper Nile is unrealistic, but you could quantify it !
• Page 9, line 33 : PCR-GLOWB has rounding errors ? That is strange and would point to numerical problems.

• Page 11, lines 15-21 : These numbers are strange. The equation in this paragraph is not balanced. Where have the missing 2km³/y gone ? Has the ground water increased or is the model not stabilized and shows different trends on the 1979-2010 period for the three configurations ? This requires some explanation.

• Section 3.2 : This section should include a discussion of the ground water recharge changes between LC1850 and LC2000 or HUM2000. This is another point where we have data to support a constructive discussions. There are many wells with over a 100 year long water table records where at least the sign of the observed recharge changes can be compared to the simulations. See for instance the study by MacDonald et al. 2016 for the Ganges.

• Page 14, line 3 : "leading to a strong increase in discharge" acknowledges better the existing relation.

• Page 14, line 24 : what supports the assertion that "crops lead to the largest reduction in evapotranspiration". Models have shown it but what data is there to support this in all generality ? Does it not depend on the crop variety, the type of agriculture (in small units or large scale) and cropping practices (number of harvests and rotations) ?

• Page 14, line 32 : How can we believe a sensitivity analysis of a model if we do not know if the model is a trustworthy reproduction of the current situation ? As I have pointed above, not only would your study be more credible by using the available observations but you could nicely qualify the simulated sensitivity.

• Figures 7 & 8 : These are really complex figures which would benefit from a more didactic presentation. Take one case to walk through the graphical representation so that your interpretation is easier to follow.

• Figure 9 : Only 1 basin seems to have significant ground water pumping as the arrow points above the actET=P line. Which basin is this and are there observations to give some credibility to this result ?

• Page 18, lines 17-21 : I think that it is important to stress that we have no way of verifying that the parameters used for pre-land-use vegetation are correct. Vegetation parameters which are used to compute evapotranspiration have been calibrated to current vegetation covers in order to obtain correct fluxes and they have no fundamental physical or biological foundation. Today's pristine forests can have functioning different from their ancestors because they are exposed to milder winters, air pollution, increased $CO_2$ levels and other environmental stresses.

• Page 18, line 24 : Is it really meaningful to distinguish between the 1850 estimated land cover and a potential cover ? I would contend that the uncertainty in the LUH data and vegetation parameters is larger than the difference to a potential cover. Could you give more substance to your hypothesis ?

• Page 19 : The discussion would be greatly helped with a table which provides the estimates of the previous studies and their main characteristics.

• Section 4.2 : I would like to restate that your have the unique opportunity to estimate the uncertainties by comparing your simulations with the observations available for most of your 100 basins. It just occurs to me that in figure 9 the century long records which exist for a number of basins could allow you to estimate the resulting arrow LC1850 → HUM2000 !

---

## Referee Comment (RC3) · Anonymous Referee #3 · 17 Feb 2017

Review of Bosmans et al "Hydrological impacts of global land cover change and human water use"

This manuscript discusses a series of global hydrologic simulations to infer the impacts of land cover change on changes in ET and subsequent water balance changes. They project the impacts this will have on discharge over major water basins. I find the manuscript clearly presented and topically appropriate for HESS. I think the conclusion that land cover change needs to be considered when studying anthropogenic impacts is important but not particularly novel as this has been shown in other regional and global studies. Nevertheless, I still think the authors make a contribution to the literature and recommend moderate revisions the manuscript at which point I think it will be suitable for publication in HESS. My major comments are below.

1. Energy balance. The authors discuss changes to ET using a model that does not contain a fully land-energy balance as many land surface models do. I think this may influence the findings of the work, particularly where the results show canceling out or reinforcement from land cover changes and the Budyko relationships. It would seem that exploration of the sensitivity (beyond what is in the SI) of this assumption on results would be important. I would like to see discussion of the impacts of the simplified approach used here contrasted with a more complete energy balance both in approach and with discussion on the impacts to the conclusions.

2. Since the authors force the model with a reference Ep (p3, lines 25+) "We force the model with CRU-TS3.21 temperature, precipitation and reference potential evapotranspiration from 1979- 2010..." and the PCRGLOB does not calculate a land energy balance on it's own, the only component that is changing within the simulation is the available water stress curve and shallow soil storage. This also would have a direct effect on the simulation results. The authors discuss the copy factor sensitivity in the SI but a discussion of the sensitivity of soil moisture storage and plant and bare soil water stress on the overall water budget and simulation results is important.

3. As I understand it, the authors compare rain-fed (p6. ~line 25) with irrigated agriculture (same page ~line 30) but do not present results for groundwater depletions. Either I'm misunderstanding the work and groundwater is not pumped in these cases or I feel there is an opportunity to present differences in abstraction with land cover change.

4. It would be interesting to compare to the simulation results to both point and remote sensing products (eg. p 19 discussion) and other studies spatially. The authors discuss total magnitudes of change but how do the spatial patterns change between model and remotely sensed inferences?

---

## Author Comment (AC1) · 31 Mar 2017

Overall comments to the editor and reviewers:
We would like to thank the reviewers for their time, their compliments on our manuscript being well written and clearly presented, as well as their valuable suggestions for improvements. Specific replies to the reviewers' comments are given per reviewer below. Here, we have copied the referees´ comments in black, with our answers in blue italics.

The main things we would like to clarify is that our study is intended as a sensitivity study, using an idealized model and forcing set-up. We did not aim for a full analysis of parameter uncertainty, which would be a topic on its own. However, the fact that different vegetation parameters may change the outcomes will be added to the Discussion. Furthermore, we will compare our experiments to discharge observations of the GRDC data base to argue for the quality of our model and experiments. Also, we will re-run our experiments. In our experiments presented in the submitted paper, multi-cropping was not included, which may explain the relatively low consumptive water use (evapotranspiration) from irrigation in experiment HUM2000. With the referees' comments, we identify this as a shortcoming and will thus address this in the revised manuscript.
* * *
**Anonymous Referee #1**

The paper describes a global assessment of hydrological impacts of land cover change and human water use for the period 1850-2000 and fits therefore well to the scope of the journal. The manuscript is well written and interesting; the figures shown in the manuscript are of good quality. Quantifying the effects of land cover change and water use on the hydrological cycle for such a long period is challenging and previous estimates varied considerably, depending on input data, models and assumptions used. Therefore, more research is needed to reduce these uncertainties. However, I think that a major revision is required before the manuscript may be considered for publication in HESS. My major points of criticism are:

General comments: 1) The authors quantify and compare the effects of land cover change and water consumption on evapotranspiration and river discharge. However, they assume that the third term in the water balance, the precipitation term, is not affected by the changes in land cover and water use (at least there is no attempt to analyze changes in precipitation). This is a strong assumption that needs at least some discussion, because the authors present here a spatial analysis. There is growing evidence in the literature that both, land cover change and water use, modify precipitation patterns over large regions (see for example Pei et al., 2016 on the effect of irrigation on summer precipitation in the US). When irrigation results in increased ET, increased ET results in increased precipitation, and increased precipitation results in increased runoff. Consequently, the net effect of irrigation on river discharge may be much smaller than the results suggested by the authors. So, the key question is certainly where water use and land cover change are taking place and in which region this will cause changes in precipitation (within the same watershed, outside of it but in an- other watershed or over the sea outside terrestrial surface). Answering this question is only possible by coupling a hydrological model with an atmospheric circulation model. This might be out of scope of the present analysis but the consequences of ignoring feedback mechanisms by changed precipitation patterns requires at least discussion.

*Author comments: indeed, the changes in land cover and particularly irrigation can affect precipitation. This feedback will be mentioned in section 4.2 where we discuss the uncertainty in input data. We wish to state that by using reanalysis data (CRU, ERA-Interim)*

*for the last 3 decades the observed changes in precipitation that reflect such a feedback are likely included in the forcing. Furthermore, to fully disentangle the effect of irrigation, both on- and offline experiments are needed. Online (coupled) experiments are indeed beyond our scope. Such experiments are typically possible within land surface models or general circulation models, which typically do not allow the inclusion of water use, dams etc as readily as PCR-GLOBWB does. Coupling PCR-GLOBWB with a global or regional climate model is daunting and beyond our resources. Our experiments are offline experiments, set up as sensitivity experiments, all being forced with the same precipitation, whether irrigation is included (HUM2000) or not (LC experiments). This allows us to focus on the direct effects of land cover change and human water use.*

2) One basic result of the study is that the effect of human water use on actual evapotranspiration is smaller than the effect of land cover change (page 11, line 6). How-ever, the increase of ET by irrigation estimated by the authors seems to be very low compared to other studies. According to the present study, global ET is increased by irrigation by 377 km3 yr-1 (page 11, line 7) while other studies reported a much larger increase in ET by irrigation (for example, > 1000 km3 yr-1 between 1900 and 2005 according to Kummu et al., 2016). Why is that? Assuming that the uncertainty in additional ET created by irrigation is that large: how would this uncertainty then affect the basic conclusions drawn by the authors?

*Author comments: our estimate of increased ET by irrigation is indeed rather low. The HUM2000 experiment did not include multi-cropping in irrigated areas, which explains (at least partly) this low value of 377 km3/yr. We realize that using multi-cropping will provide a more realistic estimate of the effect of human water use and we will include this and repeat the HUM2000 experiment for the updated manuscript.*

3) The authors explicitly pointed out that an analysis and discussion of the uncertainties involved in their estimates was not focus of the present analysis (page 14, lines 30-32). Nevertheless, these uncertainties exist and should be discussed. It is complex enough to simulate changes in ET on cropland because data for irrigated/rainfed crops and the distinction between paddy and upland crops are available for recent years only, in addition simulation of ET for the period outside the cropping season requires many assumptions. Even more complex and extremely difficult is it to estimate changes in ET caused by the use of ecosystems as pasture. There are many different types of pasture characterized by distinct species composition, different proportion of woody biomass and different stocking densities. There are very intensive types of pasture with properties very similar to cropped surfaces and extensive pasture systems that hardly differ from natural vegetation. It remains completely unclear how the authors reflected this complexity in the parameterization of their model to estimate realistic changes in ET caused by the conversion of natural vegetation to pasture. In addition, there are large uncertainties about the historical extent of pasture. Currently available data sets differ considerably in their estimates. The authors mention these uncertainties in section 4.2 but it remains unclear how much the basic results of the study are impacted by these uncertainties. How robust are the results of the study? More description and discussion is needed.

*Author comments: the effect of uncertainty in parameter values on our estimates is indeed excluded as we present an idealized sensitivity study using one model version (and hence one model parameter set). Papers such as Boisier et al (2014) show that model results may vary due to differences in land cover parameterization, different land cover maps and different evapotranspiration rates of land cover products (as referred to in section 4.2). Assessing the robustness of our results (i.e. 'how sensitive is the sensitivity to e.g. land cover change') would be another study in itself.*

*Within the PCR-GLOBWB model, changes in ET between various crop types are taken into account by basing the crop factors on the 26 crop types in MIRCA. Crop factors also vary seasonally, as does the ground cover of the land cover types, thus taking into account differences in ET in and outside the cropping season.*

*Pasture is indeed a complex land use type. Within our parameterization, we allow for spatial variation in the parameter values depending on local variations. In the crop types (rainfed crops, irrigated paddy, irrigated non-paddy) this largely reflects the abundance of the different crop types. For pasture, this reflects a mixture of actual pastures or meadows and grazed, semi-natural lands. This information is derived from the GLCC land cover types, identifying which land cover types are preferred. These types are subsequently selected locally on the basis of their presence in the GLCC coverage and the required area of pasture/rangeland in the controlling dataset (e.g., HYDE). Thus, various types of pasture/rangeland are created, selecting for example grassland in NW Europe as an equivalent of intensive dairy farming but shrublands or savannah in drier parts of the world as equivalents of more extensive, pastoral systems. This will be clarified in the Methods section of the updated paper. A table with the areas of GLCC land cover types identified as pasture, per gridcell, will be send in with this rebuttal[1]. GLCC IDs 2, 7, 10, 19, 34, 40, 41, 42, 55, 56, 57, 58, 93 and 94 can be used as pasture, see appendix 1 on https://lta.cr.usgs.gov/glcc/globdoc2_0 for a description of each land cover ID.*

4) The text section is often difficult to read because it contains too many numbers and reads to technical (e.g. section 4.1; section 3.2). I recommend to report the general findings in the text section and detailed results in tables. It may also help to develop a figure presenting the main results of the study (changes in terrestrial ET and discharge by water use and land cover change at global scale).

*Author comments: we agree, the text will be simplified by adding tables / figures representing the main numbers and findings of our study.*

Minor comments: Abstract: Please report more in detail how the present study adds to a better understanding of the impact of lands cover change and water use on terrestrial hydrology. What is reported in the second part of the abstract represents more the state of knowledge but not new findings and conclusions from the present analysis. *The abstract will be adapted as suggested.*

Page 2, lines 25-29: This sentence is hard to understand. Please simplify. *Will do.*
Page 3, lines 5-9: Please simplify. Not nice to have brackets in brackets . . . . *Will do.*
Page 3, lines 28-30: ERA-Interim and CRU data often differ considerably, in particular for precipitation and number of wet days. Is this not a problem when combining these two products? *PCR-GLOBWB requires daily meteo input as forcing and to this end we combined ERA-Interim and CRU TS 3.21 on a spatial resolution of 0.5 degrees. We included the CRU primarily to correct the rainfall depths in the reanalysis but are aware that certain areas in certain periods are not always covered by stations. Thus, we include CRU information only if stations are present in the sphere of influence of a half-degree cell for the month under consideration. So, if the CRU has matching station data in a cell, the spatial interpolated precipitation amounts are used to scale the ERA-Interim precipitation to the correct depth. To this end, we first remove drizzle (applying a threshold of 0.1 mm per day) and then the CRU monthly precipitation is proportionally apportioned to the resulting days rain days according to ERA-Interim daily. If no CRU information is available, the ERA-Interim information is used directly, redistributing the removed drizzle proportionally over the significant raindays. We*
* * *
[1] Area_table_pasture.tbl, providing per gridcell the longitude and latitude as well as the area per GLCC id.

*prefer to use the temporal rainfall distribution of the ERA-Interim over the CRU as the ERA-Interim reflects the continuous state of the atmosphere on a daily resolution whereas the CRU is statistically interpolated and the rainfall distribution is sensitive to the changing number of contributing stations over time. The number of raindays of the CRU is only used when a proportional scaling of the two precipitation datasets is not feasible, for example when one of the precipitation amounts is zero or virtually nil. In that case, raindays are added to match the number in the CRU and those days given an amount that brings the total to the observed total depth. In general, this only concerns the arid and semi-arid regions of the world and its influence on the global precipitation distribution is negligible. (see Van Beek et al., 2011 for a similar procedure using CRU and ERA-40).*

Page 3, line 33: More description is needed how the different land cover types were parameterized to account for different types of pasture vegetation and crops. For example, the rooting depth may vary considerably even within the 6 major land cover classes used by the authors. *This will be clarified, see also point 3 above. The land cover types can exist of different types of crops or vegetation based on the distribution of crops and vegetation in the GLCC and MIRCA data sets. This distribution, and thus the combination used for each land cover type, varies spatially, hence the resulting parameter values are spatially distributed. In Figures 1 and 2 at the end of this document we show the root fractions in the two soil layers as an example – showing that even within one land cover type the distribution of roots differ from cell to cell. For crops, there is little variation in the root fractions as roots typically only extend into the upper layer. For natural vegetation and pasture there is a greater range.*

Page 6, section 2.3: How were reservoirs treated in LC1850 and LC2000? *They are excluded in both of these experiments (i.e. no anthropogenic impacts on the water flow).*

Page 11, lines 6-7: "as evapotranspiration is only increased over irrigated areas". => This is an assumption made by the authors, however, in reality ET has also changed considerably in rainfed crops due to land use modification. *True, but here we refer to the comparison of experiments LC2000 and HUM2000 with have the same land cover, the only difference is that water is redistributed in HUM2000 (added to irrigated crops for instance). This will be clarified, e.g. by adding a reference to Figure 5b.*

References: Kummu, M., Guillaume, J.H.A., de Moel, H., Eisner, S., Florke, M., Porkka, M., Siebert, S., Veldkamp, T.I.E. and Ward, P.J., 2016. The world's road to water scarcity: shortage and stress in the 20th century and pathways towards sustainability. Scientific Reports, 6, 38495.

Pei, L.S., Moore, N., Zhong, S.Y., Kendall, A.D., Gao, Z.Q. and Hyndman, D.W., 2016. Effects of Irrigation on Summer Precipitation over the United States. Journal of Climate, 29, 3541-3558.
* * *
**Review #2 of "Hydrological Impacts of Global Land Cover Change and Human Water use" by Bosmans et al.**

Generally, the paper is interesting and well written. But for the moment it does not contribute originally to the literature on the impact of land use on the continental water cycle. Indeed, this topic has been studied with many land surface models. But these are all numerical experiments which trust blindly that the parameters for the various vegetation types (Which have been tuned for the current climate and vegetation distribution.) apply to the original

vegetation which existed before the human started to change landscapes massively in the mid 19 th century. The authors acknowledge only partly this fact in the discussion section of the paper.

Coming from the global hydrological models community, the authors have a trump they should use. In contrast to classical LSMs, PCR-GLOWB is designed to simulate today's water usage and thus should simulate quite realistic river discharges in current conditions. Thus, the simulation HUM2000 should be much more realistic than the simulations on which the other land use studies are based. In other words, I would expect this study to present the realism of this simulation to argue for the quality of his study and its added value. Furthermore, the use of the deviation from observed discharges could serve as an estimate of uncertainty and qualify the global averages changes in actET and discharge presented in section 4.1.

Thus, and before proposing a list of minor comments, I would suggest a major revision of this paper so as to present hydrological arguments as to why we should trust your numbers more than those of the cited papers. Else this paper will be just more noise on a topic where for the moment we are just guessing some numbers and anybody can propose "alternative facts".

*Author comments: the aim of our study is to provide a sensitivity analysis of the separate and joint impacts of land cover change and human water use on the terrestrial water cycle, in particular surface water availability, in PCR-GLOBWB. This by itself is novel. The reviewer is correct in stating that the parameterization of the vegetation types may affect the outcomes, but here we provide an idealized sensitivity analysis rather than an analysis of parameter uncertainty. The overall goal as well as the parameter uncertainty being a point of discussion will be made clearer in the updated version of the paper.*

*Previous assessments of PCR-GLOBWB's validation include, amongst others, Wada et al (2011, 2014), showing model validation against observations from sources such as GRDC (Global Runoff Data Centre), FAO Aquastat (for water use) and GRACE (for total water storage). Such studies show a good agreement for discharge and water storage in most catchments, as well as good agreement for water use per country. This will be made clearer in the methods section of the updated manuscript.*

*Here, we will take into account the suggestion to present the realism of our experiments by comparing them to discharge from the GRDC (Global Runoff Data Centre), providing this comparison for major river basins in terms of mean results such as monthly mean discharge as well as statistics such as the Kling-Gupta Efficiency (less sensitive to extreme values and biases than the Nash-Sutcliffe efficiency, e.g. Lopez et al 2017). If, as expected, the HUM2000 simulation is more realistic this indeed supports the relevance of our study and can serve as an estimate of uncertainty. We wish to stress that our simulations were set up as sensitivity studies, thus leaving out some of the details and sacrificing realism, for instance keeping land cover or water use fixed during each experiment, in order to capture the major impacts of land cover change and human water use on the terrestrial water cycle.*

Below you will find some minor comments which will hopefully help improve the paper. These comments also illustrate the major changes I would deem necessary to raise the level of this paper above the previous studies on this topic.

• Page 1, Line 21 : It is not true that few studies focus on including land use. Most land surface models used in the CMIP5 simulations apply a land use scenario. It could be true that the work is not very visible in the literature. I would attribute that to the fact that this is only a set of guestimates as the vegetation parameters are highly tuned and cannot claim to have any generality.

*The CMIP5 simulations do indeed include land use scenarios, but the focus of studies using CMIP5 simulations (typically GCMs) is not on surface water availability / terrestrial hydrology.*

*Only a handful of studies focus on the latter. The reviewer is correct that each GCM interprets the land use scenarios differently (i.e. translates the fractional crop and pasture cover into model specific parameter sets depending on model set-up, resolution etc).*

• Page 2, Lines 1-15 : In your review you do not mention that the picture is further muddied by the fact that in parallel to the land & water use change climate and aerosol loadings have evolved. Thus, potET has a significant trend through modification of incident long-wave and solar radiation, atmospheric turbulence, water vapour pressure deficit and amplitude of the diurnal cycle.
I can understand that this is outside of the scope of your study but these caveats need to be mentioned in the introduction. The literature is plentiful on this topic!

*Our aim was to perform sensitivity experiments, singling out the effects of land cover change and anthropogenic water use by keeping all other boundary conditions fixed (including model parameters and climate). Indeed, the climate around 1850 was slightly different from the present day, in the revised paper we will mention in the discussion that this may affect the results. In order to include changing climate over time longer experiments including climate change are needed (this is our next step for a future paper, using climate input from GCMs from 1850 to 2100, and we believe this present sensitivity study provides essential information to interpret those more complex results).*

• Page 3, lines 25-30 : Please state clearly that you assume that the potET estimated for the period 1979-2010 is valid in 1870. To me this casts a big shadow over all land use studies but intellectual honesty requires that this is stated as a working hypothesis !

*This will be stated (see previous comment)*

• Pages 18, line 18 : Can this impact on the Nile really be trusted ? The observed discharge in Aswan for the period 1871-1900 (i.e. before the first dam) is 112km3/y or about 3500 m3/s (What does PCR-GLOWB say?). Your combined change (HUM2000-LC1850) seems to be above 100m3/s, thus the amount of water in the Nile at Aswan should have increased! Observations indicate that the inflow into the great dam has not changed significantly since the end of the 19 th century. On the other hand, the amount arriving at the sea has dramatically been reduced. As you see, the value of your hydrological model is that you can check the reality of the predicted changes with the observations which date back to the 19 th century. Based on my own experience the land use change proposed by LUH for the upper Nile is unrealistic, but you could quantify it!

*In our experiments the Nile outflow (at the delta) is 461, 575 or 572 km3/yr (LC1850, LC2000, HUM2000), which is indeed above the observed pre-Aswan values. PCR-GLOBWB thus does not perform well in the Nile region, likely a consequence of PCR-GLOBWB being an un-calibrated model. We will mention this in the updated version of the paper and remark that results for the Nile need to be considered with caution.*

*The upstream land cover change does include large areas of pasture taking over tall natural vegetation, hence it is not strange that land cover has a strong impact on the discharge. The smaller impact of human water use may be related to multi-cropping not being included, which likely results in irrigation water consumption being too low.*

• Page 9, line 33 : PCR-GLOWB has rounding errors ? That is strange and would point to numerical problems.

*These rounding errors are in the post-processing of the PCR-GLOBWB output and thus do not point to internal numerical problems; it concerns small rounding errors in the water balance that was drafted from the model output and that does not include explicitly the change in storage among others (see below).*

• Page 11, lines 15-21 : These numbers are strange. The equation in this paragraph is not balanced. Where have the missing 2km3/y gone ? Has the ground water increased or is the model not stabilized and shows different trends on the 1979-2010 period for the three configurations ? This requires some explanation.

*The disbalance in the equation can be related to both rounding errors in the post processing as well as a remaining trend in the water storage. When including storage change into the equation, the equation becomes: dDesalinized = dQ + dET + dConsumption + dTWS, where TWS is total water storage (besides groundwater it also includes storage in the soil, canopy*

*and waterbodies). Note that 'dGWfossil' is now included in dTWS and therefore is excluded from the equation. Because of (fossil) groundwater abstraction, dTWS due to human water use is much larger than dTWS due to land cover change (see table 1 below). The remaining disbalance is on the order of a few km3/yr, related to rounding errors, small compared to for instance the global discharge (which in LC2000 is ~48,200 km3/yr). This will be updated in the new version of the paper.*

| in km3/yr | dLC (LC2000 – LC1850 | dHUM (HUM2000-LC2000) |
| --- | --- | --- |
| dDesalinized | 0 | 1.2 |
| dQ | 1058.3 | -906.8 |
| dET | -1048.4 | 532.9 |
| dConsumption | 0 | 504.6 |
| dTWS | -15.0 | -125.6 |

*Table 1: water balance terms for changes due to land cover (dLC) or human water use (dHUM) in km3/yr.*

• Section 3.2 : This section should include a discussion of the ground water recharge changes between LC1850 and LC2000 or HUM2000. This is another point where we have data to support a constructive discussions. There are many wells with over a 100 year long water table records where at least the sign of the observed recharge changes can be compared to the simulations. See for instance the study by MacDonald et al. 2016 for the Ganges.
*Indeed, such long records are available and could be compared to a transient model experiment. Here, we use sensitivity experiments, "time slices", which hinders a comparison to such records.*
• Page 14, line 3 : "leading to a strong increase in discharge" acknowledges better the existing relation.
*This will be updated in the revised paper.*
• Page 14, line 24 : what supports the assertion that "crops lead to the largest reduction in evapotranspiration". Models have shown it but what data is there to support this in all generality ? Does it not depend on the crop variety, the type of agriculture (in small units or large scale) and cropping practices (number of harvests and rotations) ?
*This is an assertion supported by our experiments (Figure 8), which show that overall ET is reduced most when crops replace natural vegetation. Our experiments do include a variety of crop types as well as irrigation (in HUM2000) (this will be made clearer in the updated version of the paper).*
• Page 14, line 32 : How can we believe a sensitivity analysis of a model if we do not know if the model is a trustworthy reproduction of the current situation ? As I have pointed above, not only would your study be more credible by using the available observations but you could nicely qualify the simulated sensitivity.
*See our comments above, a comparison to observed discharge will be made to see how trustworthy the model outcomes are. We will however also further clarify that the experiments were not set up to specifically represent the current situation as realistically as possible.*
• Figures 7 & 8 : These are really complex figures which would benefit from a more didactic presentation. Take one case to walk through the graphical representation so that your interpretation is easier to follow.
*These will be explained better in the revised paper.*
• Figure 9 : Only 1 basin seems to have significant ground water pumping as the arrow points above the actET=P line. Which basin is this and are there observations to give some credibility to this result ?
*This basin is a drainage basin in North Africa, draining into the Gulf of Sirte. This area is very dry and thus water limited, hence potET is much larger than P and actET is naturally very close to P. In experiment HUM2000 the actET/P is only slightly higher than in LC2000, in this very dry area not much water needs to be added to reach actET>P. There are other basins*

*where the impact of human water use on actET/P is much greater (see grey arrows in Figure 9). Both surface and groundwater can be a source for additional actET.*

• Page 18, lines 17-21 : I think that it is important to stress that we have no way of verifying that the parameters used for pre-land-use vegetation are correct. Vegetation parameters which are used to compute evapotranspiration have been calibrated to current vegetation covers in order to obtain correct fluxes and they have no fundamental physical or biological foundation. Today's pristine forests can have functioning different from their ancestors because they are exposed to milder winters, air pollution, increased CO2 levels and other environmental stresses.

*Indeed, we do not take into account that the 1850 vegetation parameters may have been different, this will be stated in the updated version of the paper. The vegetation parameters are derived from the GLCC and MIRCA datasets which represent present-day vegetation, but no calibration to fluxes has been done in these datasets (GLCC is based on remote sensing, MIRCA on crop statistics mostly).*

• Page 18, line 24 : Is it really meaningful to distinguish between the 1850 estimated land cover and a potential cover ? I would contend that the uncertainty in the LUH data and vegetation parameters is larger than the difference to a potential cover. Could you give more substance to your hypothesis ?

*There is a difference in the 1850 land cover according to LUH, which includes e.g. pasture and crop in western Europe (Figure 3d) and a fully potential, e.g. natural, cover. Indeed, there are differences in land cover data (e.g. HYDE vs Sage as mentioned in the discussion), but here we simply state the differences between the studies with the use of potential land cover instead of 1850 land cover as one of the potential reasons for different outcomes.*

• Page 19 : The discussion would be greatly helped with a table which provides the estimates of the previous studies and their main characteristics.

*Such a table will be provided in the updated version of the paper.*

• Section 4.2 : I would like to restate that you have the unique opportunity to estimate the uncertainties by comparing your simulations with the observations available for most of your 100 basins. It just occurs to me that in figure 9 the century long records which exist for a number of basins could allow you to estimate the resulting arrow LC1850 → HUM2000!

*We will indeed compare our simulations to GRDC discharge. In Figure 9 it will be difficult to add an estimate based on observations as there is, to our knowledge, no data source that indicates basin-wide changes in actET and potET for 1850 to present-day. Also, our experiments are set up as sensitivity experiments, complicating the comparison to such records.*
* * *
**Anonymous Referee #3**

Review of Bosmans et al "Hydrological impacts of global land cover change and human water use"

This manuscript discusses a series of global hydrologic simulations to infer the impacts of land cover change on changes in ET and subsequent water balance changes. They project the impacts this will have on discharge over major water basins. I find the manuscript clearly presented and topically appropriate for HESS. I think the conclusion that land cover change needs to be considered when studying anthropogenic impacts is important but not particularly novel as this has been shown in other regional and global studies. Nevertheless, I still think the authors make a contribution to the literature and recommend moderate revisions the manuscript at which point I think it will be suitable for publication in HESS. My major comments are below.

1. Energy balance. The authors discuss changes to ET using a model that does not contain a fully land-energy balance as many land surface models do. I think this may influence the findings of the work, particularly where the results show canceling out or reinforcement from land cover changes and the Budyko relationships. It would seem that exploration of the sensitivity (beyond what is in the SI) of this assumption on results would be important. I would like to see discussion of the impacts of the simplified approach used here contrasted with a more complete energy balance both in approach and with discussion on the impacts to the conclusions.

*Author comments: indeed, we do not compute the energy balance within the PCR-GLOBWB model, a water-balance model that uses prescribed potential reference evapotranspiration. Our experiments are thus set up as idealized sensitivity experiments, which will be clarified further in the updated version of the paper. Future work will focus on including the energy balance when investigating the land cover changes, using the model VIC which can be run both in water-balance mode as well as a full energy balance mode.*
*Results from a comparison between PCR-GLOBWB and VIC (both in water balance and full energy balance mode; WB and EB) in a master student's report indicate that while VIC-EB outperforms VIC-WB in some aspects of the water balance, the improvement due to the energy balance is small. Furthermore, PCR-GLOBWB scores are on average similar to both VIC versions. An example of comparing snow water equivalent of these models is given in Figure 3.*

2. Since the authors force the model with a reference Ep (p3, lines 25+) "We force the model with CRU-TS3.21 temperature, precipitation and reference potential evapotranspiration from 1979- 2010:" and the PCRGLOB does not calculate a land energy balance on it's own, the only component that is changing within the simulation is the available water stress curve and shallow soil storage. This also would have a direct effect on the simulation results. The authors discuss the copy factor sensitivity in the SI but a discussion of the sensitivity of soil moisture storage and plant and bare soil water stress on the overall water budget and simulation results is important.

*Author comments: the changes between our simulations are either the land cover (LC2000 vs LC1850), with land cover-specific and spatially varying parameters such as root fractions, interception capacity and crop factors, or whether human water use is included (HUM2000 vs LC2000). A different land cover thus results in a different distribution of the energy fluxes, as energy is limited by the potential evapotranspiration which itself broken up into interception evaporation (based on interception storage and whether the canopy is wet), transpiration and soil evaporation (based on crop factor, gap fraction and soil moisture). A different root depth distribution as a result of the land cover differences between LC2000 and LC1850 also acts to change the transpiration between the two experiments (see root depth distribution, spatially varying, per land cover type in Figures 1 and 2). This will be further clarified in the methods and discussion of the updated paper.*

3. As I understand it, the authors compare rain-fed (p6. line 25) with irrigated agriculture (same page line 30) but do not present results for groundwater depletions. Either I'm misunderstanding the work and groundwater is not pumped in these cases or I feel there is an opportunity to present differences in abstraction with land cover change.

*Author comments: we do indeed represent both rainfed as well as irrigated agriculture, with irrigation only being applied in the experiment HUM2000. HUM2000 is the only experiment in which groundwater is pumped for human water use (this will be clarified in the updated version of the paper). Hence there is no comparison to be made of abstraction for the LC1850 and LC2000 experiments.*

4. It would be interesting to compare to the simulation results to both point and remote sensing products (eg. p 19 discussion) and other studies spatially. The authors discuss total magnitudes of change but how do the spatial patterns change between model and remotely sensed inferences?

*Author comments: we will compare our results to observed discharge from the GRDC data base (see other reviewer comments). A comparison of the spatial patterns of change between model output and remotely sensed interferences is beyond the scope of our study. Furthermore, such a comparison would be hampered by the idealized set up of our experiments as well as the short period of availability of remotely sensed products (the time scale of those is decades whereas we for instance compare land cover of 1850 to 2000).*

References:
Lopez, P. L., Sutanudjaja, E., Schellekens, J., Sterk, G., & Bierkens, M. Calibration of a large-scale hydrological model using satellite-based soil moisture and evapotranspiration products. *Hydrology and Earth System Science Discussions, 2017*
Wada, Y., Van Beek, L., Viviroli, D., Dürr, H. H., Weingartner, R., and Bierkens, M. F.: Global monthly water stress: 2. Water demand and severity of water stress, *Water Resources Research*, 47, 2011.
Wada, Y., Wisser, D., and Bierkens, M.: Global modeling of withdrawal, allocation and consumptive use of surface water and groundwater resources, *Earth System Dynamics*, 5, 15, 2014.

[Figure]

Figure 1: Root fraction in soil layer 1 (upper soil layer, 0.13 to 0.3 m deep). The higher the root fraction the more root is in the upper layer (and thus not reaching the lower layer).

[Figure]

Figure 2: Root fraction in soil layer 2 (lower soil layer, 0.52 to 1.2 m deep). The higher the root fraction the more root is in the lower layer (and thus able to pick up moisture from both layers), a value of 0 means that the roots do not extend into the second soil layer.

[Figure]

Figure 3: Kling-Gupta Efficiency (KGE), Nash-Sutcliffe Efficiency (NSE), and correlation scores (r) for three model runs (PCR-GLOBWB, VIC-EB, VIC-WB, all forced with WFDEI data) for snow water equivalent compared to ASMR-E. Outliers are not shown in the box plots. For details see master thesis by Lars Killaars[2], "Hydrologic response modelling: comparing the VIC and PCR-GLOBWB models", February 2016.
* * *
[2] MSc thesis Utrecht University and University of Washington. See https://www.dropbox.com/s/p869jecxp1nt1ok/Killaars%20Scriptie%20Final%20version.docx?dl=0

---

## Author Comment (AC2) · 31 Mar 2017

This is the table mentioned in the response to referee #1.

Please also note the supplement to this comment:
http://www.hydrol-earth-syst-sci-discuss.net/hess-2016-621/hess-2016-621-AC2-supplement.zip